# VAT: Vision Action Transformer by Unlocking Full Representation of ViT

## Abstract

In robot learning, Vision Transformers (ViTs) are standard for visual perception, yet most methods discard valuable information by using only the final layer's features. We argue this provides an insufficient representation and propose the Vision Action Transformer (VAT), a novel architecture that is extended from ViT and unlocks the full feature hierarchy of ViT. VAT processes specialized action tokens with visual features across all transformer layers, enabling a deep and progressive fusion of perception and action generation. On a suite of simulated manipulation tasks, VAT achieves a 98.15% average success rate across four LIBERO benchmarks, establishing a new state-of-the-art by outperforming prior methods like OpenVLA-OFT. Our work presents not only a powerful model for imitation learning but also demonstrates the critical importance of leveraging the complete "representation trajectory" of vision models to advance robotic policy.

## 1 Introduction

Embodied Artificial Intelligence (Embodied AI) represents a frontier research domain that bridges artificial intelligence with robotics, emphasizing the integration of physical interaction as a core component of intelligent behavior (Kim et al., 2025; Nvidia et al., 2025). Unlike traditional AI systems that operate solely in digital spaces (Vaswani et al., 2017; DeepSeek-AI et al., 2025), embodied agents actively perceive and interact with their environments, enabling them to learn, adapt, and execute tasks in dynamic physical or simulated worlds. In Embodied AI, imitation learning has been widely explored (Zare et al., 2023; Osa et al., 2018). Human-collected robot operation data provides high-quality demonstration trajectories for deep learning models to learn.

Currently robot imitation learning predominantly adopts two training paradigms. The first paradigm focuses on training task-specific robotic policies, exemplified by architectures such as Diffusion Policy (Chi et al., 2023) or Action Chunking with Transformers (ACT) (Zhao et al., 2023). These models integrate vision encoders with policy networks to decode robotic actions from visual observations. The second paradigm, termed Vision-Language-Action (VLA) models (Kim et al., 2024; 2025; Black et al., 2024), combines visual perception capabilities with the robust generalization abilities of Large Language Models (LLMs). VLA approaches aim to achieve generalizable robotic manipulation capabilities when trained on sufficient imitation learning datasets (Padalkar et al., 2023).

Both paradigms rely fundamentally on visual perception for task execution. Leveraging Vision Transformers(ViT), such as SigLIP (Zhai et al., 2023) and DINOv2 (Oquab et al., 2023), encodes visual observations into visual features. These features subsequently condition action generation. However, a critical concern arises: features from such models may inadequately encode fine-grained visual details (e.g., object geometry or precise spatial attributes), potentially impeding environmental comprehension and compromising action precision.

This concern originates from the very nature of a ViT's feature generation process. ViT is composed of a sequence of stacked Transformer layers. During the computation of ViT, the representation of the visual input is progressively transformed and enriched at each layer, through which each layer yields a new representation derived from the previous one, ultimately culminating in a final output. The representations produced sequentially across layers can be can be conceptualized as a "representation trajectory" which reflects the model's evolving interpretation of the visual input.

During the training of a ViT, the final representation(the visual features given by the last transformer layer) is explicitly optimized by minimizing a loss function, as it is directly used to generate a prediction that is compared against the ground truth. In contrast, the intermediate representations along the trajectory are only updated indirectly via backpropagation. For instance, in SigLIP, the final representation is optimized to align with the semantic representation of a corresponding text description via a contrastive loss. This training objective enables SigLIP to achieve excellent performance on zero-shot image classification tasks when provided with suitable text templates. Similarly, in DINOv2, the final representation is optimized through a knowledge distillation loss, which aligns the student model's output with that of a teacher model. This self-supervised methodology allows DINOv2 to produce powerful, general-purpose representations well-suited for dense prediction tasks such as segmentation and depth estimation.

Despite their effectiveness, the final representations from models like SigLIP and DINOv2 exhibit critical limitations: SigLIP struggles to preserve pixel-level detail, while DINOv2 can discard local, low-level information. Conversely, representations from earlier layers often retain these valuable characteristics. This leads us to posit that relying solely on the final ViT layer provides a static and impoverished representation, discarding a wealth of information crucial for robotic tasks. Yet, the conventional paradigm in robot learning is to extract only these final-layer features. This critical oversight serves as the primary motivation for our work: to enhance robot policy performance by exploiting the entire representation trajectory of a ViT.

Guided by this motivation, we propose Vision Action Transformer (VAT), a novel robot policy model that elegantly leverages visual representations in every transformer layer of ViT to elevate the upper performance bounds of imitation learning.

Notably, to achieve VAT, we introduce a simple structural extension to the ViT architecture. We introduce additional tokens (termed action tokens) to the sequence of input vision tokens from ViT's patch embedding layer, and process the combined sequence through the VAT. During the forward computation, vision tokens are processed by the vision module, which uses the original ViT parameters, while action tokens are handled by the action module, which shares the same structure as the vision modules but is initialized with new parameters, enabling them to attend to vision tokens via cross-attention. Subsequently, the processed action tokens are projected by a lightweight action decoder head to output robot actions. Experimental validation on robot simulation benchmarks demonstrates that VAT significantly enhances robot policy capacity under limited training iterations.

Our contributions are summarized as follows:

1. We identify and validate the importance of leveraging the full hierarchy of visual representations throughout the ViT architecture for excellent performance on robot learning tasks.

2. We propose VAT, a novel and simple policy architecture. By achieving state-of-the-art performance on the LIBERO benchmark suite, we demonstrate the superior potential of VAT.

## 2 RELATED WORKS

Imitation learning is a prominent methodology for robot manipulation, enabling models to learn from expert demonstrations and control robots for specific tasks. Research in this domain has progressed along several key directions. For instance, Dasari & Gupta (2020); Duan et al. (2017); James et al. (2018) emphasize multi-task or few-shot learning, while Jang et al. (2022); Brohan et al. (2022); Shridhar et al. (2021; 2022) leverage multimodal information such as depth maps or point clouds. Still Pastor et al. (2009); Zeng et al. (2020); Johns (2021); Shridhar et al. (2022) focus on designing specialized model architectures. A representative work in this area is ACT, which introduces a simple yet effective framework implemented on the Aloha platform that enables stable training and inference. A recent trend in robot policy learning involves scaling data, a paradigm derived from the scaling laws observed in large language models (Padalkar et al., 2023; Brohan et al., 2022; Walke et al., 2023; Khazatsky et al., 2024; Kalashnikov et al., 2018). For example, Team et al. (2024) is a generalist policy trained on large-scale data that can control multiple robots out-of-the-box and supports flexible fine-tuning to new robot setups. Furthermore, many studies have explored the use of Vision-Language Models (VLMs) for robotics, directly fine-tuning large pretrained VLMs to predict robot actions (Padalkar et al., 2023; Brohan et al., 2023; AI et al., 2024; Wayve, 2024; Huang et al., 2023; Li et al., 2023b). Such models are often referred to as Vision-

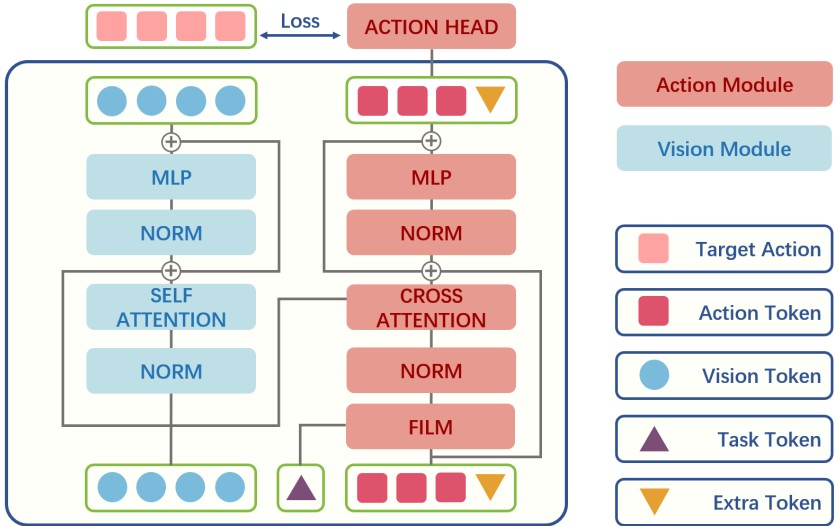

Figure 1: VAT architecture within a single layer. A standard ViT block (Vision Module, left) processes vision tokens. In parallel, a new Action Module (right) updates action tokens by cross-attention to vision tokens. Task-specific information is injected via a FiLM layer, and the action module mirrors the vision module's structure but with its own trainable parameters.

Language-Action (VLA) models, as they fuse robot control actions directly into VLM backbones. The use of a generic VLM architecture, rather than one custom-made for robot policy, allows robot policies to benefit from the rapid improvements in VLM training.

Imitation learning for robotic policies has achieved significant progress in various aspects, including data and model architecture. However, the fundamental principle remains unchanged: generating appropriate robotic actions based on visual observations. Consequently, the perception of visual scenes is a critical component of imitation learning. Current approaches typically employ ViT, such as SigLIP, or DINO (Caron et al., 2021), to extract visual representations that serve as conditional inputs for generating robot actions. A common method for acquiring these representations involves extracting the feature map from the final layer of a ViT (Karamcheti et al., 2024; Liu et al., 2023). While effective, this approach raises concerns that such a representation may not sufficiently capture all the information required for downstream robotic tasks (Lan et al., 2024).

The field of Vision-Language Models (VLMs) has recognized the limitations of single-layer features and has explored several avenues to obtain richer visual representations. One common strategy is to fuse features from multiple, distinct ViT encoders. For instance, Tong et al. (2024) utilizes a Mixture of Features to integrate visual features from CLIP-ViT (Radford et al., 2021) and DINOv2, while Lu et al. (2024) employs a hybrid vision encoder that combines SigLIP-L (Zhai et al., 2023) for low-resolution inputs and SAM-B (Kirillov et al., 2023) for high-resolution inputs. A more closely related strategy, however, is to fuse features from multiple layers within a single ViT. This multi-layer fusion typically follows two main schemes: external and internal (Lin et al., 2025). External fusion integrates features from different ViT layers before they are passed to the language model. For example, Yao et al. (2024) directly concatenates features from multiple layers, while Cao et al. (2024) uses a cross-attention module where final-layer features query shallower ones to extract fine-grained details. While effective, this approach faces a direct trade-off: richer information from more layers comes at the cost of increased computational load due to longer vision token sequences. Internal fusion, in contrast, injects multi-layer visual features at different layers within the LLM backbone (Meng et al., 2024). This avoids increasing the input sequence length, allowing models like Qwen3-VL (Qwen Team, 2025) to progressively incorporate hierarchical visual information. This principle of progressive, multi-layer integration aligns closely with the insight behind our VAT architecture. However, both external and internal fusion share a fundamental limitation: they require a heuristic or costly search process to select which layers to fuse. This arbitrary choice risks using a suboptimal set of features. Our VAT framework elegantly sidesteps this issue entirely. By design,

VAT integrates visual representations from every layer, systematically leveraging the full feature hierarchy without the need for manual layer selection or extensive ablation studies.

Another branch of VLM research involves native multi-modal models, which omit a separate ViT entirely (Diao et al., 2025; Chen et al., 2024; Diao et al., 2024). In these architectures, visual and textual features are processed jointly within a single, unified transformer, with each layer containing parameters for both modalities. However, this unified design comes at a significant cost: it forgoes the powerful representations from pre-trained ViTs, as the vision processing parameters are typically trained from scratch or shared with the language model (Mo et al., 2025; Li et al., 2023a). Nevertheless, these models offer an inspiring paradigm: the concurrent, layer-wise refinement of visual features as they interact with another modality. This contrasts sharply with traditional pipelines where a static visual representation is fully computed first and only then integrated downstream. This principle of concurrent interaction is the key insight we adapt in VAT. However, instead of discarding the ViT, VAT implements this paradigm within the vision backbone itself. It facilitates a progressive, layer-by-layer interaction between the evolving visual representation and the robot's action modality. In doing so, VAT captures the best of both worlds: the powerful, pre-trained features of a ViT and the dynamic, interactive processing of layer-wise visual features.

## 3 METHOD

We introduce VAT, an architectural advancement built upon ViT. As illustrated in Figure 1, VAT extends the standard ViT framework by integrating a specialized action module for robot learning, while retaining the original ViT's core visual representation capabilities. Specifically, within each layer, the original ViT components are kept as vision modules, and we introduce new action modules that are identical in structure but have their own randomly initialized parameters. After extension, VAT processes a concatenated sequence of vision tokens and action tokens as input. In vision module, vision token processing within each transformer layer follows this computational paradigm:

$$\mathbf{x}_{\text{vision}}' = \mathbf{x}_{\text{vision}} + \text{Attention}\big(\text{LayerNorm}_1(\mathbf{x}_{\text{vision}})\big) \tag{1}$$

$$\mathbf{x}_{\text{vision}_{\text{out}}} = \mathbf{x}_{\text{vision}}' + \text{MLP}\big(\text{LayerNorm}_2(\mathbf{x}_{\text{vision}}')\big) \tag{2}$$

Action tokens are processed by the action module, interacting with vision tokens via cross-attention mechanisms. Following the final transformer layer, the output action tokens are decoded by an action prediction head to produce executable robot actions. The decoded value of each action token corresponds to a specific dimension of an individual action within the action chunk.

VAT is optimized using an L1 loss between the predicted and target action values. The length of each action token is K × L, where: K denotes the chunk size (number of actions in an action chunk), L represents the dimensionality of each action. We set K to 8. For the LIBERO dataset L is 7, where the first six dimensions represent the delta position and rotation of the end effector, and the last dimension indicates the open / closed gripper state. In experiments, we initialize each action token as a zero vector and add trainable positional embeddings to each token. To enable the model to be aware of the task type, we assign a unique task token to each task. Within each layer of the VAT, we employ Feature-wise Linear Modulation (FiLM) to generate task-specific scaling factors from the task token. These factors are then used to modulate the action tokens. This mechanism ensures that the model is explicitly conditioned on the type of task it is currently performing. The FiLM computation process is as follows:

$$\mathbf{t}_{\text{embed}} = \text{TaskEmbeddingLayer}\,(\text{task\_id}) \tag{3}$$

$$\mathbf{\Theta}_{\text{film}} = \text{FilmModulator}\,(\mathbf{t}_{\text{embed}}) \tag{4}$$

$$\gamma,\ \beta = \text{Split}\,(\mathbf{\Theta}_{\text{film}}, \text{dim} = 2) \tag{5}$$

$$\mathbf{x}_{\text{action}} = \mathbf{x}_{\text{action}} \odot (\gamma + \mathbf{1}) + \beta \tag{6}$$

After FiLM, action token processing within each transformer layer follows the computational paradigm below:

$$\mathbf{x}_{\text{action}}' = \mathbf{x}_{\text{action}} + \\ \text{CrossAttention}\left(\text{LayerNorm}_3(\mathbf{x}_{\text{action}}), \text{LayerNorm}_1(\mathbf{x}_{\text{vision}})\right) \tag{7}$$

$$\mathbf{x}_{\text{action}_{\text{out}}} = \mathbf{x}_{\text{action}}' + \text{MLP}_{\text{action}}\left(\text{LayerNorm}_4(x_{\text{action}}')\right) \tag{8}$$

It is noteworthy that $\mathbf{x}_{\text{vision}}$ in equation 1 and equation 7 refers to the vision tokens from adjacent lower layer. This implies that the action tokens at a given layer interact with the vision tokens from the preceding layer via cross-attention. When VAT employs diffusion loss instead of L1 loss, it becomes necessary to incorporate diffusion timestep information. In the first layer of the VAT, we concatenate a diffusion timestep embedding token to the $\mathbf{x}_{\text{action}}$ token sequence. This allows the action tokens to acquire timestep information during cross-attention. However, since $\mathbf{x}_{\text{vision}}$ in equation 7 serves as the query and key, the resulting output $\mathbf{x}_{\text{action}}'$ from the cross-attention operation does not retain the timestep token. To ensure the diffusion timestep information propagates explicitly through all layers, we also concatenate the timestep token to the $\mathbf{x}_{\text{action}}$ token sequence in the first layer. As a result, the timestep token remains present in the $\mathbf{x}_{\text{action}}$ after cross-attention. The same approach is applied for incorporating robot proprioceptive information: we project the proprioception data into a token and concatenate it with $\mathbf{x}_{\text{action}}$. These tokens are referred as extra tokens in Figure 1. Through this approach, we properly integrate essential information required during training into the computational process of VAT.

## 4    Experiments

To comprehensively evaluate the performance of VAT, we conduct experiments on simulated benchmarks. We selected LIBERO, which consists of four sub-benchmarks, each comprising 10 distinct tasks. We first performed a series of experiments to determine the optimal training configurations for VAT. Subsequently, our results on LIBERO demonstrate that VAT achieves state-of-the-art success rates, outperforming other VLA models without requiring pre-training on robot data.

### 4.1    Experimental Configuration of VAT

From the numerous transformer-based vision foundation models available, we select SigLIP 2 and DINOv2 to serve as the backbone for our VAT. We select the LIBERO benchmark to evaluate VAT's performance on a suite of robot manipulation tasks. LIBERO is composed of four distinct sub-benchmarks, each designed to test a specific capability: LIBERO-Spatial assesses the ability to generalize to new spatial arrangements of objects; LIBERO-Object evaluates the transfer of manipulation skills across visually distinct but functionally similar objects; LIBERO-Goal tests the adaptation of behaviors and action sequences to achieve different outcomes; and LIBERO-10 measures the ability to execute long-horizon tasks.

We adopt the following training configuration as our default setup. Unless stated otherwise, all experiments use these settings. Specifically, we employ a cosine learning rate scheduler with an initial rate of 2e-5 and a global batch size of 128, distributed across 4 NVIDIA A100 GPUs. All model parameters are set as trainable. For input, we utilize both the third-person and wrist camera views from the LIBERO dataset. The L1 loss function is selected for optimization. Ablation studies analyzing the impact of camera views, loss functions, and other configurations are provided in appendix A.

We follow a rigorous evaluation protocol. For each LIBERO benchmark (Spatial, Goal, Object, and 10), we train VAT for 100 epochs, saving a checkpoint every 10 epochs (yielding 10 checkpoints). Consistent with the OpenVLA-OFT protocol, we evaluate each checkpoint over 500 episodes and report the success rate of the best-performing checkpoint. The comparative results of VAT against other models on the LIBERO benchmark are presented in Table 1.

### 4.2    Model Layer Skipping

The key innovation of our VAT is its ability to leverage visual features from every layer of the ViT to enhance the robot policy. This allows the policy to perceive visual input through a variety of repre-

Table 1: Comparison of different models

|  | **Spatial** | **Object** | **Goal** | **10** | **Average Scores** |
|---|---|---|---|---|---|
| **Diffusion Policy (scratch)** | 78.3 | 92.5 | 68.3 | 50.5 | 72.4 |
| **Octo (fine-tuned)** | 78.9 | 85.7 | 84.6 | 51.1 | 75.1 |
| **DiT Policy (fine-tuned)** | 84.2 | 96.3 | 85.4 | 63.8 | 82.4 |
| **$\pi$0 (fine-tuned)** | 96.8 | 98.8 | 95.8 | 85.2 | 94.2 |
| **OpenVLA-OFT (fine-tuned)** | 97.6 | 98.4 | **97.9** | 94.5 | 97.1 |
| **VAT (default setup)** | **98.8** | **99.4** | 97.6 | **96.8** | **98.15** |

The scores for models in Table 3 except our VAT are cited directly from Kim et al. (2025). "Scratch" refers to training the Diffusion Policy solely on the LIBERO dataset. "Fine-tuned" indicates that the models are initialized with pre-trained weights and then fine-tuned on LIBERO.

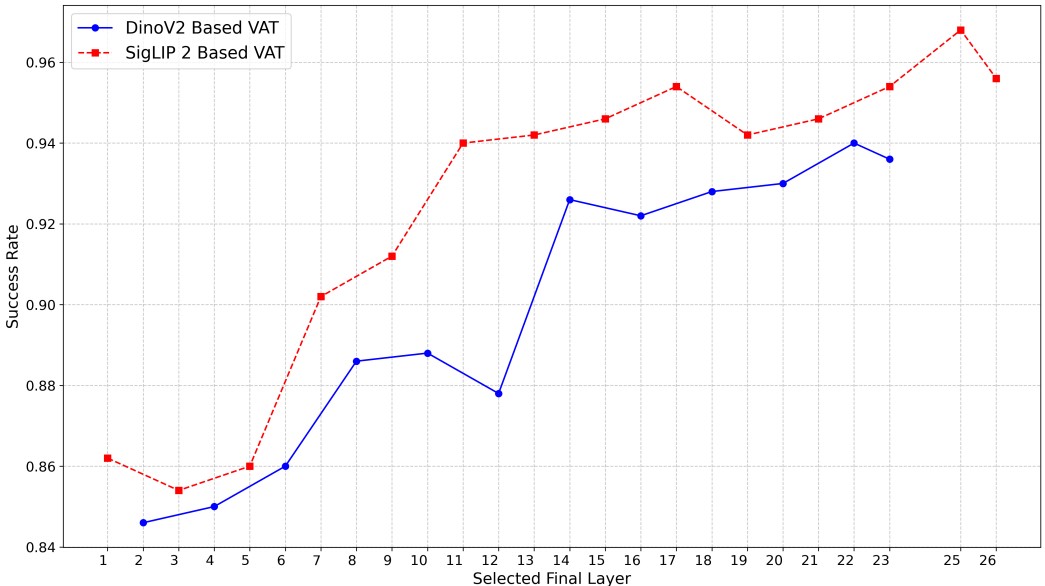

Figure 2: Results of VAT Layer Skipping Experiments

sentations, which capture both fine-grained details and high-level semantics. And a natural question is: does the policy require holistic visual features from all layers to maximize performance, or could skipping the features from later layers offer a better balance between efficiency and robustness?

To investigate this, we conduct layer-skipping experiments using SigLIP 2 and DINOv2 as backbones. Specifically, for a selected final layer, we extract the action tokens in this layer and feed them directly into the action decoder head. In this configuration, the policy only benefits from visual features up to and including the selected final layer. Figure 1 compares the performance of VAT when using different final layers, with experiments conducted on LIBERO-10 using SigLIP 2 and DINOv2 as backbones. The results demonstrate that selecting a deeper layer as the final layer tends to yield better performance for VAT in learning robot policies, which can be attributed to the fact that deeper layers enable VAT to perceive richer visual features. However, even when very shallow layers are selected, the model still achieves success rates exceeding 85% while reducing the training time by 5 to 10 times. This remarkable performance indicates that the visual features extracted from shallow layers of ViT—including the first layer—already provide sufficiently informative representations for VAT to interpret robot observations and generate actions. Thus, through this layer-skipping experiment, we further confirm that features from different layers of ViT all possess representation that could enhance robot policy learning.

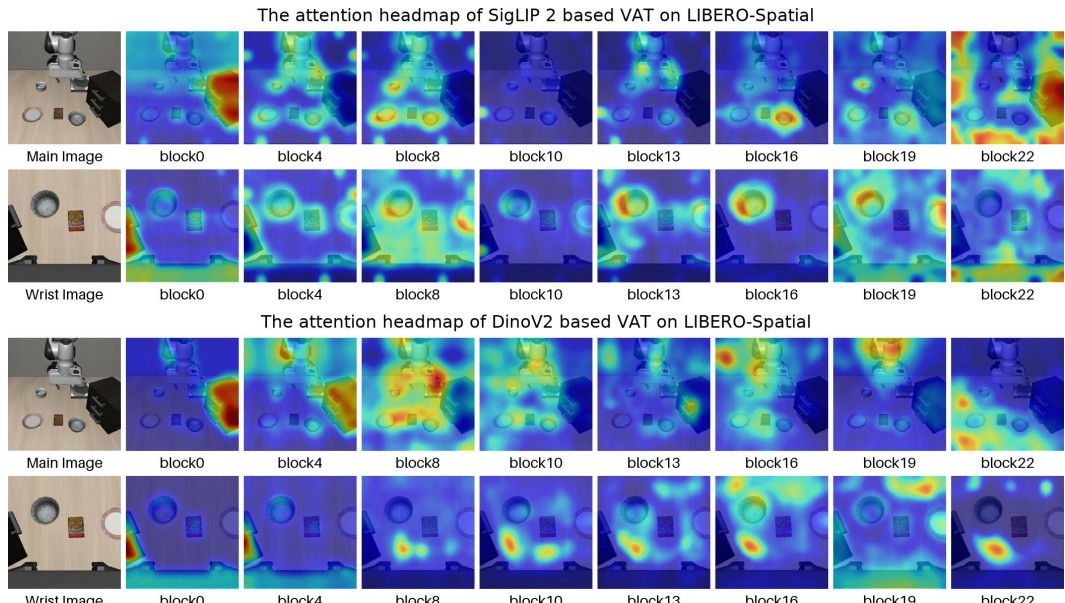

Figure 3: Attention heatmap of VAT on LIBERO-Spatial

## 4.3 ATTENTION HEATMAP VISUALIZATION

We claim that the visual features from different layers of a ViT exhibit distinct properties, all of which are beneficial for policy learning. To visually illustrate these differences, we visualize the attention scores of multiple layers as heatmaps. Specifically, we extract the attention scores where action tokens serve as the query and vision tokens as the key. This process yields an attention score tensor of shape [H, N, L] for each layer, where H represents the number of attention heads, N represents the number of action tokens and L denotes the number of vision tokens, with each vision token corresponding to a patch in the image. To derive a single attention value for each patch, we average the attention scores from all attention heads and action tokens corresponding to that patch. These averaged scores are then used to generate a heatmap, where higher values correspond to lighter colors. For enhanced visual clarity, we apply bicubic interpolation to upscale the heatmap and overlay to the original image. These heatmaps reveal the underlying mechanisms of token interaction and information flow during the model's forward computation, explicitly revealing the patterns by which it interprets and processes data.

Figure 3 visualizes the attention flow within the SigLIP 2 and DinoV2 based VAT. The heatmap of SigLIP 2 illustrates a clear "focus-then-disperse" pattern. VAT initially focus distributedly across the scene, but as information propagates throught the network, its focus sharpens and converges on the key object, and then disperses to a more global view in final layer. The heatmap of DinoV2 Exhibites a remarkably different focus: VAT demonstrates a tendency for its attention to "sink" into the background, focusing on task-irrelevant tokens. This highlights the significant representational discrepancy of DinoV2 and SigLIP 2, which in turn affects how VAT interprets visual information and process action tokens. More visualizations are presented in Appendix B.

## 4.4 COMPREHENSIVE ABLATION AND GENERALIZATION ANALYSIS

To rigorously analyze the impact of various architectural design choices and validate the robustness of our approach, we conduct a comprehensive set of ablation studies in this section. To ensure a fair and consistent comparison, unless otherwise specified, all following experiments utilize SigLIP 2 as the visual backbone, employ L1 loss for optimization, and leverage visual observations from both camera views.

**Necessity of Full Hierarchy vs. Last Layer.** A core premise of VAT is that intermediate "representation trajectories" contain critical information lost in the final layer. To validate this, we train

Table 2: Comparison of VAT and baseline methods

|  | Spatial | Object | Goal | 10 | Average |
|---|---|---|---|---|---|
| **VAT** | 98.8 | 99.4 | 97.6 | 96.8 | 98.15 |
| **baseline** | 99.2 | 94.2 | 98.2 | 74.6 | 91.55 |

Table 3: Ablation on Task Conditioning Mechanisms (%)

|  | Spatial | Object | Goal | 10 | Average |
|---|---|---|---|---|---|
| **VAT** | 98.8 | 99.4 | 97.6 | 96.8 | 98.15 |
| **No FiLM (No Task Info)** | 89.8 | 99.4 | 8.4 | 83.8 | 70.35 |
| **Task embedding** | 98.2 | 99.2 | 97.4 | 93.4 | 97.05 |

a baseline model that uses visual features exclusively from the second-to-last layer of the Vision Transformer (ViT) backbone—consistent with the approach of Kim et al. (2025)—while keeping the action module unchanged. As a result, in every layer of VAT, the action tokens perform cross-attention with visual features drawn from the second-to-last layer of the ViT. As shown in Table 2 , the Last-Layer baseline suffers a significant performance drop (98.15% → 91.55%). This degradation is most pronounced in the long-horizon LIBERO-10 benchmark (96.8% → 74.6%), confirming that the rich geometric and spatial details preserved in intermediate layers are essential for complex, multi-stage reasoning.

**Role of Task Conditioning (FiLM).** We analyze the impact of our task conditioning mechanism in Table 3. Removing task information entirely ("No FiLM") leads to catastrophic failure on Goal-conditioned tasks (8.4%), confirming that task IDs are a prerequisite for disambiguation, not a shortcut. Furthermore, replacing FiLM with a simple learnable "Task Embedding" (added to action tokens) still yields a high success rate of 97.05%. This demonstrates that while FiLM (98.15%) is the optimal design, VAT's performance is primarily driven by its hierarchical architecture rather than the specific conditioning method.

**Robustness to Action Token Capacity.** We investigate whether the number of action tokens imposes an information bottleneck on policy learning. In our default VAT configuration, consistent with Kim et al. (2025), we define an action chunk size of $K = 8$. Each action within this chunk is allocated 7 distinct tokens (representing the 6-DoF end-effector pose and gripper state), resulting in a total sequence length of 56 action tokens (8 actions × 7 tokens). To evaluate the model's sensitivity to this design, we conduct an ablation where we reduce the allocation to 3 tokens and finally to 1 token per action, while maintaining the action chunk size unchanged.As shown in Table 6, VAT exhibits remarkable stability despite this aggressive reduction. Even when compressed to a single token per action (reducing the total sequence from 56 to 8 tokens), the model achieves a 97.50% success rate. The result in Table 4 indicates that our hierarchical cross-attention mechanism is highly efficient at aggregating necessary visual cues into a compact representation, and the model is not strictly bottlenecked by the granularity of the action token sequence.

**Architecture Variants and Parameter Space Analysis.** To better understand the architectural mechanism of VAT, it is worth noting the structural parallel between our Action Tokens and the CLS token used in standard ViT training (e.g., CLIP or DINO ). Both serve as designated agents for global representation aggregation and act as the direct recipients of supervisory signals. In VAT, action tokens aggregate hierarchical visual information during the forward pass and are directly supervised by the ground-truth actions. This structural similarity reinforces the rationality of VAT's design.

However, a critical distinction lies in the Action Module. Unlike a standard CLS token that shares parameters with the visual backbone, VAT's Action Tokens are processed by a parallel Action Module—mirrored from the ViT but with independent trainable parameters. This design ensures a dedicated parameter space specifically for learning action-relevant feature extraction, preventing interference with the visual backbone's primary representation. To investigate the necessity of this dedi-

Table 4: Ablation on Action Token Number

|  | Spatial | Object | Goal | 10 | Average |
| --- | --- | --- | --- | --- | --- |
| **VAT** | 98.8 | 99.4 | 97.6 | 96.8 | 98.15 |
| **Token number 3** | 99.0 | 99.4 | 98.2 | 95.4 | 98.00 |
| **Token number 1** | 98.8 | 98.6 | 98.2 | 94.6 | 97.50 |

Table 5: Model Architecture Variants Ablation

|  | Spatial | Object | Goal | 10 | Average |
| --- | --- | --- | --- | --- | --- |
| **VAT** | 98.8 | 99.4 | 97.6 | 96.8 | 98.15 |
| **VAT-small** | 98.2 | 97.8 | 97.0 | 93.8 | 96.7 |
| **VAT-ViT** | 99 | 99.6 | 97.2 | 92.4 | 97.05 |
| **$\pi 0$** | 96.8 | 98.8 | 95.8 | 85.2 | 94.2 |
| **OpenVLA-OFT** | 97.6 | 98.4 | 97.9 | 94.5 | 97.1 |

cated parameter space and the impact of the Action Module's capacity, we evaluate two architectural variants:

VAT-Small (490M): VAT has 1.3B parameters. For VAT-Small, We reduce the dimensionality of the action tokens to one-quarter of the vision tokens. Consequently, the dimensions of the FiLM, attention, and MLP layers within the Action Module are proportionally reduced.

VAT-ViT (430M): We remove the separate Action Module entirely. In this setup, action tokens are inserted into the ViT backbone and processed using shared weights (identical to how CLS tokens are handled), utilizing simple Task Embeddings instead of FiLM.

The results in Table 5 reveal that while the full VAT (with the separate Action Module) achieves the optimal performance (98.15%), the VAT-ViT variant—which relies entirely on shared weights akin to a standard CLS token approach—still maintains a remarkably high success rate of 97.05%. This confirms two key findings:

Validity of Design: The core performance gain stems from the hierarchical access to "representation trajectories" (as both variants significantly outperform the Last-Layer Baseline in Table 4), rather than simply increasing parameter count.

Role of Separate Module: While a shared-weight approach (VAT-ViT) is viable, the dedicated parameter space provided by the separate Action Module yields better performance on complex, long-horizon tasks (LIBERO-10: 96.8% vs. 92.4%), justifying the additional architectural overhead for maximizing capability.

**Generalization on RoboTwin Benchmark.** Finally, to assess generalizability beyond LIBERO, we evaluated VAT on the RoboTwin Benchmark, which consists of 50 diverse bimanual (dual-arm) manipulation tasks. As summarized in Table 8, VAT achieves a 40.66% success rate, significantly outperforming widely used baselines such as ACT (+10.9%) and Diffusion Policy (+12.6%), and remaining competitive with the state-of-the-art VLA model Pi0 (46.42%), despite utilizing a significantly smaller backbone (1.3B vs 3B). The detailed results are shown in C.

## 5 CONCLUSION

In this work, we address a critical limitation in current robot learning paradigms: the underutilization of rich and hierarchical features from ViT. We argue that relying solely on the final layer's output provides an incomplete visual representation, potentially hindering robot policy capabilities. To overcome this, we introduce the Vision Action Transformer (VAT), a novel and parameter-efficient architecture that unlocks the full representational power of a ViT. By processing specialized action tokens alongside vision tokens through every layer of the transformer, VAT facilitates a continuous

and deep fusion of perception and action generation. This approach allows the policy to leverage the entire "representation trajectory" from the fine-grained details in early layers to the high-level semantic information in deeper ones. Our experiments on the LIBERO benchmark suite validate the effectiveness of our approach. VAT achieves a state-of-the-art success rate of 98.15% on average across four sub-benchmarks, outperforming strong baselines like OpenVLA-OFT. Furthermore, our layer-skipping analysis confirms that even shallow-layer features provide highly informative representations for policy learning, while attention visualizations offer qualitative insights into how VAT dynamically shifts its focus throughout the network. Ultimately, this work contributes not only a powerful new model for imitation learning but also a fundamental insight: fully leveraging the hierarchical features of vision models is crucial for advancing robotic perception and control. We believe that this principle will inspire future architectures in embodied AI.

## 6 ETHICS STATEMENT

This work adheres to the ICLR Code of Ethics. In this study, no human subjects or animal experimentation was involved. All datasets used were sourced in compliance with relevant usage guidelines, ensuring no violation of privacy. We have taken care to avoid any biases or discriminatory outcomes in our research process. No personally identifiable information was used, and no experiments were conducted that could raise privacy or security concerns. We are committed to maintaining transparency and integrity throughout the research process.

## 7 REPRODUCIBILITY STATEMENT

We have made every effort to ensure that the results presented in this paper are reproducible. All code and datasets have been made publicly available in an anonymous repository to facilitate replication and verification. The experimental setup, including training steps, model configurations, and hardware details, is described in detail in the paper. We have also provided the full description to assist others in reproducing our experiments.

Additionally, robot learning datasets we have used, are publicly available, ensuring consistent and reproducible evaluation results.

We believe these measures will enable other researchers to reproduce our work and further advance the field.

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

Table 6: Comparison of task performance with different camera views

| View | Spatial | Object | Goal | 10 | Average Scores |
|---|---|---|---|---|---|
| Both views (SigLIP2) | 98.8 | 99.4 | 97.6 | 96.8 | 98.15 |
| Third-person view only (SigLIP2) | 95.6 | 96.6 | 95 | 84.4 | 92.9 |
| Both views (DINOv2) | 98.2 | 99.6 | 97 | 94 | 97.2 |
| Third-person view only (DINOv2) | 96.4 | 97.8 | 96 | 84.4 | 93.65 |

Table 7: Comparison of L1 and diffusion loss

| Loss | Spatial | Object | Goal | 10 | Average Scores |
|---|---|---|---|---|---|
| L1 Loss (SigLIP2) | 98.8 | 99.4 | 97.6 | 96.8 | 98.15 |
| Diffusion Loss (SigLIP2) | 99 | 99.2 | 97.2 | 91.4 | 96.7 |
| L1 Loss (DINOv2) | 98.2 | 99.6 | 97 | 94 | 97.2 |
| Diffusion Loss (DINOv2) | 98.8 | 98.8 | 96.4 | 91 | 96.25 |
| L1 Loss(SigLIP2 cotraining) | 99.4 | 99.6 | 97.6 | 96.2 | 98.2 |
| Diffusion Loss(SigLIP2 cotraining) | 98.8 | 99.4 | 98 | 94.4 | 97.65 |

Xiaohua Zhai, Basil Mustafa, Alexander Kolesnikov, and Lucas Beyer. Sigmoid loss for language image pre-training. *2023 IEEE/CVF International Conference on Computer Vision (ICCV)*, pp. 11941–11952, 2023. URL https://api.semanticscholar.org/CorpusID: 257767223.

Tony Zhao, Vikash Kumar, Sergey Levine, and Chelsea Finn. Learning fine-grained bimanual manipulation with low-cost hardware. *ArXiv*, abs/2304.13705, 2023. URL https://api. semanticscholar.org/CorpusID:258331658.

## A  EXPERIMENTS ON TRAINING CONFIGURATIONS

### A.1  EXPERIMENTS ON CAMERA VIEWS

As shown in Table 6, the performance between using both camera views (third-person and wrist) and only the third-person camera view is compared, with using both views demonstrating superior task performance for VAT. We therefore utilize both camera views in all LIBERO experiments.

### A.2  EXPERIMENTS ON LOSS FUNCTION

As shown in Table 7, the performance between L1 loss and diffusion loss is compared, with L1 loss providing sufficient robustness. Accordingly, we select the L1 loss as VAT's loss function. We further investigate the efficacy of VAT within a co-training paradigm. By training a single VAT model on all four LIBERO subtasks, the results presented in Table 7 demonstrate its strong capacity for learning a broader range of robot policies.

### A.3  EXPERIMENTS ON TRAINABLE PARAMETERS

As shown in Table 8, the performance between training all parameters and freezing the ViT parameters is comparable, with training all parameters showing a slight edge. We therefore keep all parameters trainable during training.

### A.4  COMPARISION ON LEARNING RATE

As shown in Table 9, we try different learning rate for our VAT on LIBERO-10. results show that setting learning rate to 2e-5 is suitable for training.

Table 8: Comparison on Trainable Parameters

| View | Spatial | Object | Goal | 10 | Average Scores |
|------|---------|--------|------|-----|----------------|
| Train all parameters | 98.8 | 99.4 | 97.6 | 96.8 | 98.15 |
| Freeze ViT parameters | 98.8 | 98.6 | 97.8 | 89.8 | 96.25 |

Table 9: Comparison on Learning Rate

| Learning rate | 2e-5 | 5e-5 | 1e-4 |
|---------------|------|------|------|
| LIBERO-10 | 96.8 | 94.4 | 92.4 |

## B  VISUALIZATIONS OF ATTENTION HEATMAP

Figure 4 provides comprehensive examples that illustrate the attention flow in both SigLip 2-based and DinoV2-based VAT. In the early layers of the VAT, the heatmaps outline clear object contours from the original images. This suggests that the vision tokens in these layers, corresponding to different objects or regions, contain distinctive features. As a result, the attention patterns of action tokens toward vision tokens exhibit a strong correlation with specific objects. In contrast, in the deeper layers of VAT, the attention pattern exhibits irregular attention sink phenomena. This suggests a shift in the characteristics of visual representation, where, after extensive computation, visual information accumulates preferentially on certain background tokens. The heatmaps reveal the attention patterns present in the VAT, providing evidence for how VAT leverages visual representations from different layers to support robot policy learning.

## C  RESULTS ON ROBOTWIN BENCHMARK

To further assess the generalizability of VAT on complex, dual-arm manipulation tasks, we extend our evaluation to the RoboTwin benchmark. For the VAT implementation, we set the action chunk size to 25 and train the model for 100 epochs on each individual task, maintaining all other hyperparameters consistent with our default configuration. RoboTwin provides four camera views, so we use main camera views and views from left and right arms. For computational efficiency, we report the performance of the final checkpoint. Following the standard RoboTwin protocol, all policies are trained on the Aloha-AgileX embodiment utilizing 50 demo clean demonstrations per task. We report the success rate averaged over 100 evaluation episodes under the demo clean (Easy) setting. The results of the other models are obtained from Chen et al. (2025). The performances of VAT and other models are shown in Table 10.

## D  LLM USAGE

Large Language Models (LLMs) were used to aid in the writing and polishing of the manuscript. Specifically, we used an LLM to assist in refining the language, improving readability, and ensuring clarity in various sections of the paper. The model helped with tasks such as sentence rephrasing, grammar checking, and enhancing the overall flow of the text.

It is important to note that the LLM was not involved in the ideation, research methodology, or experimental design. All research concepts, ideas, and analyses were developed and conducted by the authors. The contributions of the LLM were solely focused on improving the linguistic quality of the paper, with no involvement in the scientific content or data analysis.

The authors take full responsibility for the content of the manuscript, including any text generated or polished by the LLM. We have ensured that the LLM-generated text adheres to ethical guidelines and does not contribute to plagiarism or scientific misconduct.

Table 10: Performance Comparison on RoboTwin

| Task | RDT | Pi0 | ACT | DP | VAT |
|---|---|---|---|---|---|
| Adjust Bottle | 81% | 90% | **97%** | **97%** | 91% |
| Beat Block Hammer | **77%** | 43% | 56% | 42% | 16% |
| Blocks Ranking RGB | 3% | **19%** | 1% | 0% | 0% |
| Blocks Ranking Size | 0% | 7% | 0% | 1% | **31%** |
| Click Alarmclock | 61% | 63% | 32% | 61% | **95%** |
| Click Bell | 80% | 44% | 58% | 54% | **94%** |
| Dump Bin Bigbin | 64% | **83%** | 68% | 49% | 72% |
| Grab Roller | 74% | 96% | 94% | **98%** | 96% |
| Handover Block | **45%** | **45%** | 42% | 10% | 0% |
| Handover Mic | 90% | **98%** | 85% | 53% | 92% |
| Hanging Mug | **23%** | 11% | 7% | 8% | 16% |
| Lift Pot | 72% | 84% | 88% | 39% | **93%** |
| Move Can Pot | 25% | **58%** | 22% | 39% | 57% |
| Move Pillbottle Pad | 8% | **21%** | 0% | 1% | 10% |
| Move Playingcard Away | 43% | 53% | 36% | 47% | **85%** |
| Move Stapler Pad | **2%** | 0% | 0% | 1% | 1% |
| Open Laptop | 59% | **85%** | 56% | 49% | 84% |
| Open Microwave | 37% | 80% | **86%** | 5% | 30% |
| Pick Diverse Bottles | 2% | **27%** | 7% | 6% | 14% |
| Pick Dual Bottles | 42% | **57%** | 31% | 24% | 25% |
| Place A2B Left | 3% | **31%** | 1% | 2% | 12% |
| Place A2B Right | 1% | **27%** | 0% | 13% | 9% |
| Place Bread Basket | 10% | **17%** | 6% | 14% | 11% |
| Place Bread Skillet | 5% | **23%** | 7% | 11% | **23%** |
| Place Burger Fries | 50% | **80%** | 49% | 72% | 66% |
| Place Can Basket | 19% | **41%** | 1% | 18% | 1% |
| Place Cans Plasticbox | 6% | 34% | 16% | **40%** | 32% |
| Place Container Plate | 78% | **88%** | 72% | 41% | 82% |
| Place Dual Shoes | 4% | **15%** | 9% | 8% | 10% |
| Place Empty Cup | **56%** | 37% | 61% | 37% | 14% |
| Place Fan | 12% | **20%** | 1% | 3% | 17% |
| Place Mouse Pad | 1% | **7%** | 0% | 0% | 3% |
| Place Object Basket | 33% | 16% | 15% | 15% | **48%** |
| Place Object Scale | 1% | **10%** | 0% | 1% | 5% |
| Place Object Stand | 15% | **36%** | 1% | 22% | 28% |
| Place Phone Stand | 15% | **35%** | 2% | 13% | 30% |
| Place Shoe | 35% | 28% | 5% | 23% | **49%** |
| Press Stapler | 41% | **62%** | 31% | 6% | 48% |
| Put Bottles Dustbin | 21% | **54%** | 27% | 22% | 39% |
| Put Object Cabinet | 33% | **68%** | 15% | 42% | 28% |
| Rotate QRcode | 50% | **68%** | 1% | 13% | 28% |
| Scan Object | 4% | **18%** | 2% | 9% | 9% |
| Shake Bottle Horizontally | 84% | **99%** | 63% | 59% | 89% |
| Shake Bottle | 74% | **97%** | 74% | 65% | 93% |
| Stack Blocks Three | 2% | **17%** | 0% | 0% | 0% |
| Stack Blocks Two | 21% | 42% | 25% | 7% | **69%** |
| Stack Bowls Three | 51% | 66% | 48% | **63%** | 51% |
| Stack Bowls Two | 76% | **91%** | 82% | 61% | 59% |
| Stamp Seal | 1% | 3% | 2% | 2% | **30%** |
| Turn Switch | 35% | 27% | 5% | 36% | **48%** |
| Average | 34.50% | **46.42%** | 29.74% | 28.04% | 40.66% |

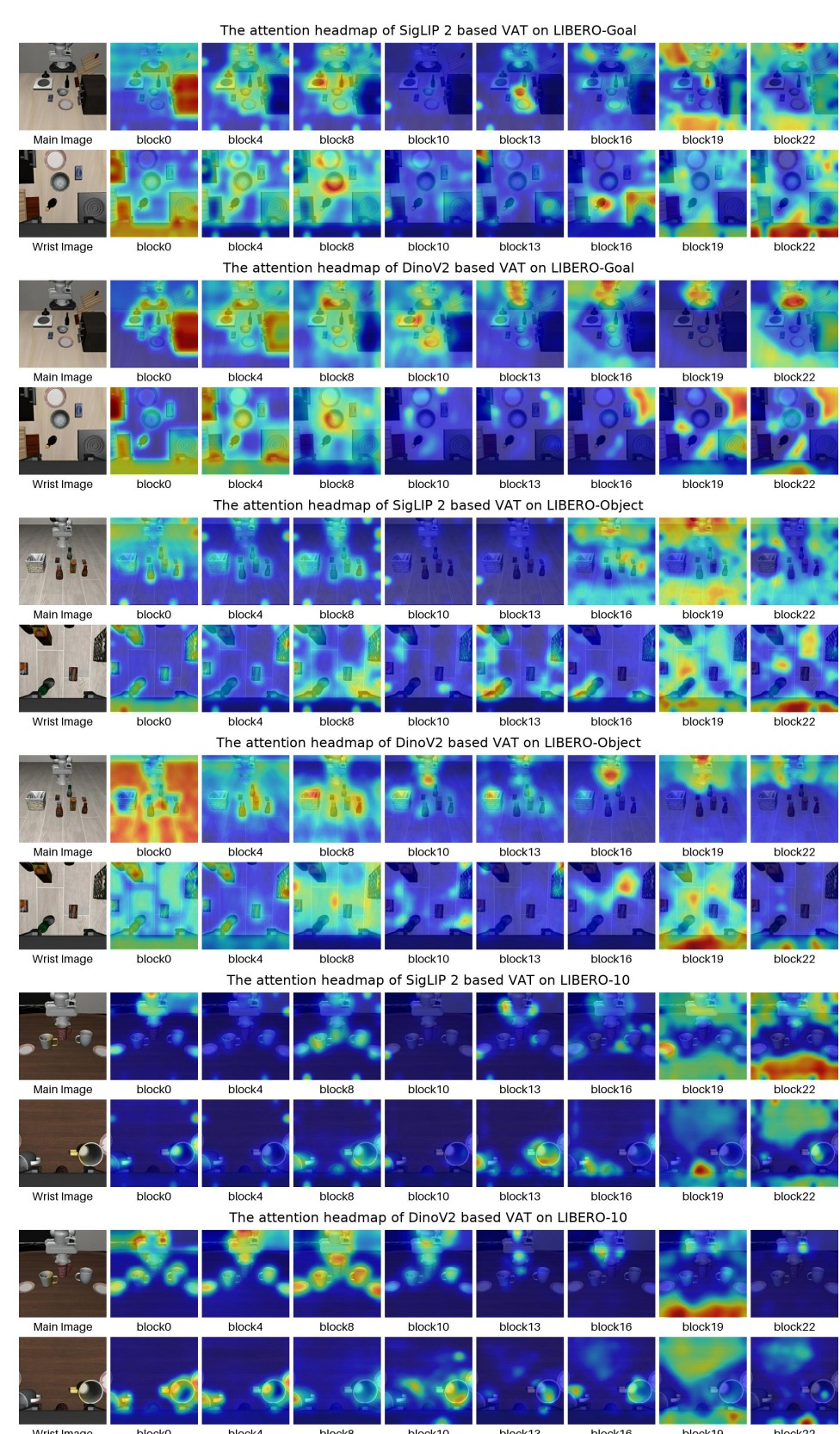

Figure 4: Attention heatmap of VAT on LIBERO-Object, Goal and 10

