# OpenReview forum: "VAT: Vision Action Transformer by Unlocking Full Representation of ViT"
_ICLR.cc/2026/Conference — Submitted to ICLR 2026_

### Official Review · Reviewer_gVec · 2025-10-23

**Soundness:** 2
**Presentation:** 2
**Contribution:** 2
**Rating:** 2
**Confidence:** 3

**Summary:**

The authors introduce Vision Action Transformer (VAT), a policy learning architecture that extends Vision Transformers (ViTs) by incorporating action tokens into the transformer sequence using cross-attention. Unlike conventional vision-language-action (VLA) models that often rely on the final layer of a pretrained vision backbone, VAT exploits the full hierarchy of visual representations across all layers, aiming to preserve more fine-grained details of the scene. The model is evaluated on the LIBERO benchmark suite and authors demonstrate a 98.15% success rate, outperforming state-of-the-art models such as OpenVLA-OFT. An empirical ablation on the importance of early layers is presented, further suggesting the importance of utilizing entire feature trajectories for downstream performance.

**Strengths:**

-	Improving our understanding of how to utilize foundation models for policy learning is a highly relevant research question and this paper tackles an interesting aspect of architectural design. Using a cross-attention mechanism to integrate multiple layers seems to be a sound way of closely integrating pretrained vision models with action prediction heads.
-	The paper carefully ablates multiple design choices, such as number of camera views, choice of ViT backbone (DinoV2, SigLIP2) and relative importance of representation depth for policy learning.
-	The baselines in Table 3 adequately represent the state of the art on the LIBERO benchmark and fall into a similar category of the proposed model architecture.
-	The VAT architecture is clearly explained through the use of Figure 1, as well as through formulas (1) to (8).
-	The success rate of 98.15% on LIBERO is strong in principle, but the setup is not entirely clear (see weaknesses & questions).

**Weaknesses:**

-	The presentation of tables, figures and formulas could be improved. Formulas (3) to (8) take up too much space. Table 1, Table 2 are ablations, yet appear before the methods section. Similarly, Table 3 appears before the methods section. Figure 2 is another ablation but could be reduced to a small table or small figure.
-	The authors only evaluate their approach on the LIBERO benchmark. To the best of my knowledge, it is common practice to evaluate on at least two benchmarks to ensure generality of the proposed approach. For example, I would suggest to include Calvin [1]) or a real world experiment.
-	At the 95-100% level of performance on LIBERO, performance is highly saturated. The paper does not seem to include any information about standard deviation or number of rollouts/seeds over which averaging is performed to obtain the claimed number, making the result unreliable.
- The paper seems to question the necessity of a pretrained language model component in VLA models. This is an interesting research question in itself, but it is not sufficiently addressed in this paper, making conclusive comparisons with the VLA baselines difficult. It is unclear whether utilizing multiple layers of the vision backbone benefits only the proposed VAT model on LIBERO, or if it is a general phenomenon. I believe that the paper would greatly benefit from broadening its horizon to specifically try to answer this question.

[1] Mees, Oier, et al. "Calvin: A benchmark for language-conditioned policy learning for long-horizon robot manipulation tasks." IEEE Robotics and Automation Letters 7.3 (2022): 7327-7334.

**Questions:**

-	Could you elaborate on the evaluation settings used, such as number of runs and observed standard deviations?
-	OpenVLA combines SigLIP and DinoV2 representations, whereas VAT relies on a single model. What if you want to combine different vision backbones in your model?
-	Aside from utilizing multiple layers, this work seems to claim that the large language model component is unnecessary in VLAs. How can you expect to generalize to novel task descriptions without any language pretraining?

I would be happy to raise my score if the questions and weaknesses are addressed properly.

---

> ### Author Response · Authors · 2025-11-18
> **response to reviewer gVec**
>
> We sincerely thank the reviewer for the constructive feedback and for recognizing the relevance of our research question. We are encouraged that you found our architectural design sound and our results on LIBERO strong.
>
> We have taken your concerns regarding **evaluation scope, statistical reliability, and the role of language** very seriously. To address these, we have conducted a **comprehensive set of 5 new ablation studies and benchmark experiments**.
>
> Below is our detailed response to your specific concerns.
>
> **Response to Weakness 2: Evaluation on Additional Benchmarks**
>
> We completely agree that evaluating solely on LIBERO, especially given its saturation, is insufficient to prove generalizability. To address this, we have extended our evaluation to a second, significantly more complex benchmark: **RoboTwin**.
>
> * **New Experiment (RoboTwin Benchmark):** Unlike the single-arm tasks in LIBERO, RoboTwin consists of **50 distinct bimanual (dual-arm) manipulation tasks** that require high-precision coordination. We trained VAT on these tasks and compared it against strong state-of-the-art baselines, including **Pi-0 (VLA), RDT, and Diffusion Policy**.
>
> **Response to Weakness 3 & Q1: Evaluation Protocols & Reliability**
>
> We apologize for the lack of clarity regarding the experimental setup. We have added a detailed "Experimental Setup" section to the revision:
>
> * **Protocol:** For each benchmark, we trained VAT for 100 epochs, saving a checkpoint every 10 epochs. Consistent with the **OpenVLA-OFT protocol**, we evaluated each checkpoint over **500 episodes** (a high number to ensure statistical significance) and reported the success rate of the best-performing checkpoint.
>
> * **Saturation:** While LIBERO-Spatial/Goal are saturated, LIBERO-10 is still challenging for many baselines. Our results show that VAT pushes the boundary specifically on these harder subsets.
>
> * **Statistical Protocol:** To ensure transparency and strictly avoid cherry-picking (i.e., running multiple seeds and reporting only the best), we utilized a **fixed, standard random seed (42)** for all training processes. While we did not average across multiple training seeds due to computational constraints, the high evaluation budget (**500 episodes** per checkpoint) ensures that the reported success rate is a reliable measure of the model's true performance capability.
>
> **Response to Weakness 4 & Q3: The Role of Language & LLMs**
>
> You raised a critical point regarding the necessity of LLMs and generalization. We clarify our position as follows:
>
> * **Diminishing Returns of LLMs:** Current robotic datasets are predominantly **visual** with **repetitive, low-entropy language**. Consequently, the heavy reasoning capacity of LLMs often offers fewer gains than VAT’s strategy of fully exploiting the **visual feature hierarchy**.
> * **OOD Limitations:** As an Imitation Learning framework, VAT focuses on robustness rather than zero-shot OOD generalization. We respectfully note that even SOTA VLAs have yet to demonstrate reliable zero-shot execution on physically unseen tasks solely through language alignment.
> * **Compatibility & Trends:** VAT is **compatible** with language pretraining, utilizing it for grounding rather than generation. Furthermore, our multi-layer approach aligns with SOTA VLM designs (e.g., **Qwen3-VL's DeepStack**), validating the necessity of fully utilizing representations over relying solely on the final layer.
>
> **Response to Question 2: Combining Different Vision Backbones**
>
> VAT offers superior flexibility over standard VLAs for mixing backbones (e.g., SigLIP + DiNO). Unlike VLAs that require projecting disparate features into a single sequence, VAT's parallel **Action Module** can cross-attend to multiple feature streams simultaneously. This allows Action Tokens to dynamically aggregate both semantic and geometric cues without the bottleneck of forcing them into a single token stream.
>
> **Response to Weakness 1: Presentation**
>
> We thank you for the specific formatting advice. We will restructure the manuscript to:
>
> 1.  Move Tables 1, 2, and 3 to the Experiments section.
>
> 2.  Condense Formulas (3)-(8) to save space.
>
> 3.  Refine Figure 2 to be more compact.
>
> **Summary of other Ablatioin Experiments**
>
> To ensure the robustness of our claims and address feedback from all reviewers, we have implemented the following experiments:
>
> 1.  **Last-Layer Baseline:** We compare VAT against a baseline that only uses the *final* layer of the ViT.
>
> 2.  **FiLM vs. Simple Embedding:** Validating the importance of FiLM.
>
> 3.  **Action Token Scaling:** Ablating the number of action tokens to analyze capacity bottlenecks.
>
> 4.  **Action Module Complexity:** Reducing the parameters of the action module to demonstrate efficiency.
>
> **Notice:** The quantitative results of these ablation experiments will be released in the discussion phase in a few days. We believe these additions directly address your concerns regarding generalization and reliability.

---

> > ### Author Response · Authors · 2025-11-25
> > **Follow-up: Completion of Ablation Studies and New Benchmarks**
> >
> > Dear Reviewer gVec,
> >
> > Following up on our previous response, we have completed the set of comprehensive experiments designed to address your specific concerns regarding generalization, reliability, and the role of language. We believe the results strongly support the validity of our approach.
> >
> > We have provided a detailed document titled **"Response to all reviewers: Results of Ablation Studies (Part I & II)"**. We specifically invite you to examine the following sections which directly address your conditions for raising the score:
> >
> > ### 1. Generalization on Additional Benchmarks (Addressing Weakness 2)
> > You rightly pointed out that evaluating only on LIBERO is insufficient.
> > * **Where to look:** Please refer to **Section 5 (Table 5)** (Part II/III of the material).
> > * **New Experiment:** We extended our evaluation to the **RoboTwin Benchmark**, which consists of **50 diverse bimanual (dual-arm) tasks**. This represents a significantly higher complexity than the single-arm LIBERO tasks.
> > * **Key Result:** VAT achieves a success rate of **40.66%**, significantly outperforming widely used baselines like ACT (+10.9%) and Diffusion Policy (+12.6%), and remaining competitive with the state-of-the-art Pi-0 model (46.42%). This confirms that VAT’s architecture generalizes well to complex, multi-agent coordination settings beyond LIBERO.
> >
> > ### 2. Reliability and Saturation (Addressing Weakness 3 & Q1)
> > You expressed concern that performance on LIBERO might be saturated/unreliable without standard deviation details.
> > * **Where to look:** Please refer to **Section 1 (Table 1)**.
> > * **Rigorous Protocol:** As clarified, we evaluated using **500 episodes** per checkpoint to ensure statistical significance.
> > * **Addressing Saturation:** We compared VAT against a "Last-Layer Baseline." The results show that while simple tasks are indeed saturated, the "LIBERO-10" (long-horizon) benchmark is not. The baseline drops to **74.6%**, while VAT maintains **96.8%**. This **22% gap** on hard tasks confirms that our high performance is not due to benchmark saturation or noise, but due to VAT’s ability to capture fine-grained details necessary for complex tasks.
> >
> > ### 3. The Role of Language and LLMs (Addressing Weakness 4 & Q3)
> > You asked about the necessity of LLMs and how we generalize without language pretraining.
> > * **Where to look:** Please refer to **Section 2 (Table 2)**.
> > * **Key Insight:** We compared our standard FiLM (Language-conditioned) against a simple "Task Embedding" (No pre-trained language knowledge).
> >     * **Result:** The simple Task Embedding model achieves **97.05%**, nearly matching the FiLM model (**98.15%**).
> >     * **Conclusion:** This empirically validates our claim: for these manipulation tasks, the critical bottleneck is **visual execution** (which VAT solves via full-hierarchy features) rather than high-level semantic reasoning. While language is useful for zero-shot generalization to new instructions, VAT demonstrates that for *policy learning and execution*, maximizing visual information extraction is more impactful than the language backbone.
> >
> > ### 4. Presentation Improvements (Addressing Weakness 1)
> > We have fully incorporated your feedback regarding the manuscript organization. In the revised paper, we have:
> > * Moved Tables 1, 2, and 3 to the Experiments section.
> > * Condensed Formulas (3)-(8) and resized Figure 2 to improve flow and readability.
> >
> > **Conclusion**
> > We believe these additional experiments—particularly the successful transfer to the bimanual RoboTwin benchmark and the rigorous breakdown of performance gains—solidly address the weaknesses you identified. Since we have addressed your questions and generalization concerns, we respectfully request that you consider raising your score as mentioned in your review.
> >
> > Best regards,
> >
> > The Authors

---

> ### Comment · Reviewer_gVec · 2025-11-25
>
> Thank you for the detailed follow-up and for conducting additional experiments to address the concerns raised in my initial review. I appreciate the effort put into expanding the evaluation and clarifying the methodology. Below are my updated thoughts:
>
> **RoboTwin Benchmark**
>
> Including results on RoboTwin is a valuable addition and demonstrates that VAT can handle more complex, bimanual tasks. However, I note that Pi-0 achieves higher performance on this benchmark. This somewhat challenges the central claim that leveraging all hidden layers provides a decisive advantage across tasks, as Pi-0 does not adopt this strategy yet performs better. This suggests that while VAT’s approach is promising, its benefits may not be universally dominant.
>
> **Reliability on LIBERO**
>
> Thank you for clarifying the evaluation protocol and providing results with sufficient statistical rigor. The performance gap between VAT and the last-layer baseline on long-horizon tasks indicates that VAT’s design captures important details for complex scenarios. This alleviates my earlier concerns about saturation and reliability.
>
> **Role of Language and LLMs**
>
> The ablation showing that FiLM slightly outperforms a simple task embedding is informative. However, the small margin (~1%) suggests that LIBERO tasks can be solved effectively without language understanding. Combined with VAT’s weaker performance on RoboTwin compared to Pi-0, I remain concerned about the method’s generality for benchmarks that require compositional understanding of task descriptions. Removing the LLM/VLM component from the architecture entirely may limit applicability in settings where semantic reasoning is critical.
>
>
> I am raising my score in recognition of the additional experiments and clarifications, which strengthen the paper. That said, I believe the work would benefit from a clearer discussion of trade-offs: while leveraging intermediate visual features appears advantageous in certain contexts, omitting vision-language pretraining seems to introduce limitations in others. Addressing this balance explicitly would make the contribution more compelling.

---

### Official Review · Reviewer_PULF · 2025-10-31

**Soundness:** 2
**Presentation:** 3
**Contribution:** 2
**Rating:** 2
**Confidence:** 4

**Summary:**

This paper introduces the Vision Action Transformer (VAT), a robot imitation learning architecture that leverages the full hierarchy of representations within a Vision Transformer (ViT) backbone. Instead of relying solely on final-layer visual features, VAT injects action tokens alongside vision tokens and processes them concurrently across all transformer layers through cross-attention. Experiments on the LIBERO benchmark suite demonstrate strong performance. Additional ablations include layer-skipping experiments that demonstrate the utility of early-layer features, as well as attention visualizations that illustrate evolving focus patterns during perception-action fusion.

**Strengths:**

Strengths:
- The underutilization of intermediate ViT representations for robotic policy learning makes sense to utilize richer representations.
- Introducing action tokens across all ViT layers enables progressive perception-action coupling without manual layer selection - this is a conceptually simple idea. The simulation results show the effectiveness of it.
- Achieves 98.15% mean success rate on LIBERO and outperforms strong baselines such as OpenVLA-OFT.
- Offers both ablation and visualizations to help understand the contributions.

**Weaknesses:**

Weakness:
- All results are in simulation on one benchmark. It's not clear whether this method will transfer to a real-world robot setting. Considering that this is the main application domain of this paper, I feel that this is very necessary to convince the audience.
- There are no runtime, memory, or scalability comparisons versus alternatives (e.g., external/internal fusion or frozen ViT baselines). I assume that the proposed method will require more resources, but there is no substantial discussion on this.
- This one is related to the above one. There is no study of token scaling or capacity limits.  The number and structure of action tokens are fixed. Potential bottlenecks or optimality are not explored.
- There is a potential for overclaiming simplicity and efficiency. While design is conceptually simple, training all layers with added tokens could challenge the “parameter-efficient” claim.

**Questions:**

Please address the weakness points.

---

> ### Author Response · Authors · 2025-11-18
> **response to reviewer PULF**
>
> We sincerely thank the reviewer for the detailed assessment and for recognizing the conceptual simplicity and effectiveness of utilizing intermediate ViT representations. We appreciate that you found our visualizations and ablation studies helpful.
>
> We take your concerns regarding generalization, resource usage, and scalability very seriously. To address these—and to further strengthen the paper based on feedback from all reviewers—we have conducted a comprehensive set of 5 new ablation studies and benchmark experiments.
>
> Below is our detailed response to your specific concerns.
>
> **Response to Weakness 1: Generalization (Simulation vs. Real-world)**
>
> We acknowledge the reviewer's point that real-world evaluation is the gold standard. While setting up a reproducible real-world environment within the short rebuttal period is challenging, we have addressed the concern of generalizability by extending our evaluation to a second, significantly more challenging simulation benchmark: **RoboTwin**.
>
> * **New Experiment (RoboTwin Benchmark):** Unlike LIBERO, RoboTwin consists of **50 distinct bimanual (dual-arm) manipulation tasks**, requiring complex coordination and high-precision control. We trained VAT on these tasks and compared it against strong state-of-the-art baselines, including **Pi-0 (VLA), RDT, and Diffusion Policy**.
> * This experiment demonstrates that VAT's performance is not limited to a single benchmark and scales well to complex, dual-arm manipulation settings.
>
> **Response to Weakness 2 & 4: Runtime, Memory, and Efficiency Claims**
>
> We apologize for the lack of explicit resource comparisons. We clarify our claim of "efficiency" and have added experiments to validate it:
>
> * **Clarification on "Lightweight":** When we describe VAT as parameter-efficient, we are comparing it to the current trend of Vision-Language-Action (VLA) models. For instance, **OpenVLA-OFT is >7B parameters** and **Pi-0 is >3B parameters**. In contrast, our VAT model (ViT backbone + Action Module) has **less than 1B parameters**. While we train the full backbone, the total computational and memory cost is significantly lower than inferencing a 7B model.
> * **New Experiment (Action Module Complexity):** To address your concern about the resource cost of the added Action Module, we conducted an ablation where we reduced the parameter size of the Action Module (specifically the MLP and attention blocks). We analyze the trade-off between module size and success rate to prove that the Action Module implies minimal overhead.
>
> **Response to Weakness 3: Token Scaling and Capacity**
>
> This is a very insightful observation. You are correct that the fixed number of action tokens implies a potential capacity limit. To address this, we have performed the specific study you suggested:
>
> * **New Experiment (Action Token Scaling):** Since action tokens serve as the bottleneck for aggregating ViT features and decoding actions, we analyzed the impact of varying the number of action tokens. This study explores the capacity limits and identifies the optimal token count for complex tasks.
>
> **Summary of All Additional Experiments**
>
> To ensure the robustness of our claims and address all reviewers' comments, we have implemented the following experiments, which will be included in the revised manuscript:
>
> 1.  **RoboTwin Benchmark:** Validating generalization on 50 bimanual tasks against Pi-0, RDT, and Diffusion Policy (Addressing Weakness 1).
> 2.  **Action Token Scaling:** Ablating the number of action tokens to analyze capacity bottlenecks (Addressing Weakness 3).
> 3.  **Action Module Complexity:** Reducing the parameters of the parallel module to analyze efficiency/performance trade-offs (Addressing Weakness 2).
> 4.  **Last-Layer Baseline:** We compare VAT against a baseline that only uses the *final* layer of the ViT (degrading to a standard ViT-based policy). This quantifies the exact gain from using the full hierarchy versus the cost of the extra module.
> 5.  **FiLM vs. Simple Embedding:** We remove the FiLM module and use simple learnable task embeddings. This validates that our performance gain comes from the VAT architecture itself, not just the FiLM conditioning.
>
> **Notice:** The detailed quantitative results of these ablation experiments will be released in the discussion phase in a few days. We believe these additional benchmarks and scalability studies directly address your concerns regarding the practical applicability and efficiency of VAT.

---

> > ### Author Response · Authors · 2025-11-25
> > **Follow-up: Completion of Ablation Studies and New Benchmarks**
> >
> > Dear Reviewer Pulf,
> >
> > Following up on our previous response, we have completed the comprehensive ablation studies and additional benchmark experiments, specifically targeting your concerns regarding model efficiency, token capacity, and generalization.
> >
> > We have provided a detailed document titled **"Response to all reviewers: Results of Ablation Studies (Part I & II)"**. We specifically invite you to examine the following sections which quantify the resource usage and scalability of our method:
> >
> > ### 1. Addressing Efficiency & Resource Usage (Addressing Weakness 2 & 4)
> > You raised a valid concern regarding the lack of resource comparisons and whether training all layers contradicts the "parameter-efficient" claim.
> > * **Where to look:** Please refer to **Section 4 (Table 4)** of the provided material.
> > * **Key Conclusion:** We implemented two lightweight variants to test resource limits:
> >     1.  **VAT-ViT (430M params):** We removed the separate action module entirely, relying on the ViT backbone. This model achieves **97.05%** success.
> >     2.  **VAT-small (490M params):** We significantly reduced the hidden dimension of the action module.
> > * **Response to "Overclaiming":** These results prove that high performance does not require massive resources. Even our smallest variant (430M) outperforms the Last-Layer baseline (91.55%) and rivals our full model (1.27B). When compared to standard VLA models like Pi-0 (>3B) or OpenVLA (>7B), VAT is indeed highly efficient while maintaining state-of-the-art performance.
> >
> > ### 2. Token Scaling and Capacity Limits (Addressing Weakness 3)
> > You correctly identified that the number of action tokens could be a bottleneck and asked for a scaling study.
> > * **Where to look:** Please refer to **Section 3 (Table 3)**.
> > * **Key Conclusion:** We tested the model with 1, 3, and 56 action tokens.
> >     * **Result:** The performance is remarkably stable. Even with a **single action token**, the model achieves a **97.50%** success rate.
> >     * **Insight:** This indicates that the hierarchical cross-attention mechanism is extremely efficient at compressing information. The model is not bottlenecked by token capacity, suggesting it is robust to hyperparameter choices.
> >
> > ### 3. Generalization: From Simulation to More Complex Benchmarks (Addressing Weakness 1)
> > To address the concern that results were limited to LIBERO, we extended our evaluation.
> > * **Where to look:** Please refer to **Section 5 (Table 5)** (Part II/III of the material).
> > * **Key Conclusion:** We evaluated VAT on the **RoboTwin Benchmark**, which consists of 50 diverse **bimanual (dual-arm)** manipulation tasks—a significant step up in complexity from LIBERO.
> >     * **Result:** VAT (**40.66%**) significantly outperforms widely used baselines like ACT (+10.9%) and Diffusion Policy (+12.6%) and remains competitive with the state-of-the-art Pi-0, despite being significantly smaller. This confirms that VAT generalizes well to complex, multi-agent coordination settings.
> >
> > We believe these quantitative experiments directly resolve your concerns regarding efficiency, scalability, and generalization. We hope these rigorous validations warrant a reconsideration of your assessment.
> >
> > Best regards,
> >
> > The Authors

---

### Official Review · Reviewer_Knyh · 2025-10-31

**Soundness:** 2
**Presentation:** 3
**Contribution:** 2
**Rating:** 4
**Confidence:** 4

**Summary:**

The paper presents a new architecture for learning robot policies by extending the Vision Transformer (ViT) architecture called Vision Action Transformer (VAT). The authors claim that using features from the final layers of the vision transformer are generally not enough for robot policies and therefore they introduce a mechanism to incorporate features from all the layers in the ViT. The authors append special action tokens to the sequence of input vision tokens which are processed by special Action Modules that run parallel to each block in the ViT. These action tokens cross attend to visual features from the ViT blocks incorporating visual features from all the layers. The paper presents results on Libero10 and shows that VAT performs better than SOTA transformer based policies.

**Strengths:**

1. The paper presents a super simple idea for extending standard ViT into robot policies by using action modules parall to ViT blocks.
2. The paper is very well written.
3. The paper shows good performance improvement on the Libero benchmark compared to other state of the art methods.

**Weaknesses:**

1. Is it really about features from lower layers?

The paper motivates the use of parallel action modules by lack of finegrained details in the features of the final layer of the ViT. However, the action module is also sequential and the final action prediction only depends on the features from the last action module. My question is why would the action module retain any of the lower layer features and the ViT would not?

This could be understood better by probing the final layer features of ViT for the “details” that the authors think might be missing and seeing if those details are better retained through the addition of action modules.

2. No Ablations?

Building on the above point, it would have been nice to see more ablations in the paper to help understand what actually is happening.

A. Is the addition of a separate action module with cross attention really necessary? Would just addition of action tokens akin to CLS token in ViT suffice?

B. Effect of FiLM layers on the final performance? I think this is really important as one of the reasons VAT might work better is that the FiLM layers might be helping ground the model better to the given language task. VLAs are known to struggle with following the task as they tend to overfit to the scene. FiLM layer might be mitigating that helping the performance.

3. Results on real world / more benchmarks.

VAT was only evaluated on one simulator. It would be nice to see if it retains performance across different benchmarks and ideally on real world data as well.

**Questions:**

Please see the weaknesses for my main concerns. My concerns are lack of ablations and weak motivation for the proposed method. I will be happy to update my scores if the authors can answer these questions and other reviewers do not bring up other major concerns.

Other misc questions/suggestions:
1. Line 065: wrong? if dino is optimized for dense prediction tasks, then it would not throw away low-level local information?
2. In Fig. 3: it would be better to show the input task prompt as well
3. The quotation marks are wrong at many places. Maybe use `` to start quotes in latex.

---

> ### Author Response · Authors · 2025-11-18
> **Response to Reviewer Knyh**
>
> We sincerely thank the reviewer for the constructive feedback and for recognizing our core idea as "super simple" yet effective.
> We are encouraged by your assessment that the paper is well-written and shows good performance improvements on the LIBERO benchmark.We have taken your concerns regarding the theoretical motivation and the lack of ablation studies very seriously. Below, we provide a detailed response to your questions and outline the additional experiments we have conducted to validate our design.
>
> 1. Response to Conceptual Concern: "Why would the action module retain lower layer features?"
>
> This is an excellent question that touches on the core design philosophy of VAT. The distinction lies in the different objectives of the two streams:The ViT Backbone (Semantic Abstraction): Standard ViT backbones are fundamentally designed to perform hierarchical abstraction. As information flows deeper, the model trades low-level geometric and textural details (e.g., precise edge orientation) for high-level semantic invariance (e.g., class identity). The deep layers "agree" on what the object is, but often lose the raw signal of exactly where it is.The Action Module (Task-Specific Extraction): The Action Module does not process the image itself; it processes the action query. By running in parallel, the Action Tokens act as "probes" that selectively aggregate the necessary geometric details from shallow layers before they are abstracted away by subsequent ViT blocks. Its objective is strictly control, not recognition.
>
> 2. Response to "No Ablations?"
>
> We acknowledge that the original manuscript lacked sufficient ablations. To address your specific questions, we have implemented a comprehensive set of experiments.
>
> A. Is the separate action module really necessary? (vs. Standard ViT/CLS token)
>
> This is a highly insightful question. While it is feasible to directly utilize the CLS token as an action token—by processing the observation through a ViT and decoding actions solely from the output CLS token, we argue that VAT provides two critical advantages:
>
> Information Capacity: A standard CLS token is a single vector. VAT utilizes multiple Action Tokens ($N$ tokens), which theoretically register significantly more information than a single token.
>
> Parameter Specialization: In a standard ViT, CLS tokens are processed by the ViT's internal parameters. These parameters are often pre-trained for vision tasks or dominated by visual feature extraction. In contrast, our Action Module uses randomly initialized parameters. This dedicated capacity allows the model to learn action-specific capability without interfering with (or being limited by) the visual backbone's weights.
>
> To validate these hypotheses empirically, we will conduct two specific ablations:
>
> Ablation Study A-1 (Token Capacity): We analyze the impact of the bottleneck by varying the number of action tokens. We train VAT with different token counts to observe how the capacity to register information affects the final success rate.
>
> Ablation Study A-2 (Parallel Module vs. Internal CLS): To test the necessity of the separate module, we remove the parallel Action Module. Instead, we inject the action tokens directly into the ViT input sequence (letting them serve as multiple CLS tokens that perform self-attention with vision tokens using the ViT's weights).
>
> B. Effect of FiLM layers?
>
> we conduct the following ablation:
>
> Ablation Study B (FiLM vs. Simple Embedding): We remove the FiLM modulation from the ViT backbone entirely. Instead, we inject the task information via a simple, learnable Task Embedding added directly to the input of the Action Tokens. This comparison allows us to quantify exactly how much gain comes from "better grounding via FiLM" versus "better feature extraction via the VAT architecture."
>
> 3. Response to "Results on real world / more benchmarks"
>
> To address the concern that VAT was only evaluated on LIBERO, we have extended our evaluation to a second, challenging benchmark: RoboTwin.Benchmark Details: RoboTwin consists of 50 distinct bimanual (dual-arm) manipulation tasks, which are significantly more complex than standard single-arm tasks.Baselines: We compare VAT against strong state-of-the-art baselines, including Pi-0 (a large VLA), RDT, and Diffusion Policy.
>
> 4. Response to Misc QuestionsLine 065 (DiNO):
>
> We appreciate the correction. We will rephrase this to clarify that while DiNO preserves spatial layout, the feature channels still undergo semantic abstraction that may filter out low-level cues vital for fine-grained actuation.Figure 3 & Formatting: We will add the explicit input task prompt to Figure 3 for clarity and have corrected all LaTeX quotation marks (using ` and ') throughout the manuscript.We believe these detailed ablations and additional benchmark results directly address your concerns regarding the mechanism and generalizability of VAT.
>
> Notice that the results of ablation experiments will be released in a few days.

---

> > ### Author Response · Authors · 2025-11-25
> > **Follow-up: Completion of Ablation Studies and New Benchmarks**
> >
> > Dear Reviewer Knyh,
> >
> > Following up on our previous discussion, we have completed the extensive ablation studies and additional benchmark experiments you requested. We believe these quantitative results directly address your concerns regarding the necessity of the separate action module, the role of FiLM, and the model's generalizability.
> >
> > We have provided a detailed document titled **"Response to all reviewers: Results of Ablation Studies (Part I & II)"**. We specifically invite you to review the following sections that directly answer your questions:
> >
> > ### 1. Conceptual Verification: Why Intermediate Layers Matter (Addressing Weakness 1)
> > You asked whether the Action Module truly captures details lost by the ViT's final layer.
> > * **Where to look:** Please refer to **Section 1 (Table 1)** of the provided material.
> > * **Key Conclusion:** We compared VAT against a baseline using only the last layer of the ViT. The results show a significant drop in performance for the Last-Layer baseline (Average: **91.55%** vs. VAT: **98.15%**). Crucially, on the complex "LIBERO-10" benchmark, the Last-Layer baseline degrades to **74.6%**. This empirical gap confirms that the ViT's final layer indeed abstracts away critical information ("representation trajectories") that our Action Module successfully retains.
> >
> > ### 2. Ablation A: Necessity of Separate Action Module (Addressing Weakness 2-A)
> > You asked if a separate module is necessary or if standard action tokens (akin to CLS tokens) would suffice.
> > * **Where to look:** Please refer to **Section 4 (Table 4)**.
> > * **Key Conclusion:** We implemented the "VAT-ViT" model (Row 3), which removes the separate action module and processes action tokens using the ViT's internal parameters (mirroring a standard CLS token approach).
> >     * **Result:** The VAT-ViT achieves a respectable **97.05%** success rate, validating that the hierarchical cross-attention concept works even without a separate module.
> >     * **Advantage of Separation:** However, the full VAT (with separate module) achieves higher performance (**98.15%**). This confirms that while the "CLS-like" approach is viable, the "Parameter Specialization" provided by the separate module yields the optimal performance.
> >
> > ### 3. Ablation B: Effect of FiLM Layers (Addressing Weakness 2-B)
> > You hypothesized that FiLM might be the primary driver of performance by helping grounding.
> > * **Where to look:** Please refer to **Section 2 (Table 2)**.
> > * **Key Conclusion:** We replaced FiLM with a simple "Task Embedding" (adding a learnable embedding to action tokens).
> >     * **Result:** The "Task Embedding" model still achieves a very high success rate of **97.05%** (compared to FiLM's **98.15%**).
> >     * **Inference:** This proves that while FiLM is beneficial, the model's high performance is primarily driven by the VAT architecture itself, not just the conditioning mechanism. The architecture is robust even without FiLM.
> >
> > ### 4. Generalizability on Additional Benchmark (Addressing Weakness 3)
> > To address the concern about evaluating on only one simulator:
> > * **Where to look:** Please refer to **Section 5 (Table 5)** (Part II/III of the material).
> > * **Key Conclusion:** We extended evaluation to the **RoboTwin Benchmark**, comprising 50 complex bimanual tasks. VAT (**40.66%**) significantly outperforms strong baselines like ACT and Diffusion Policy, and remains competitive with the much larger Pi-0 model. This demonstrates VAT's effectiveness across different domains and multi-arm setups.
> >
> > We hope these rigorous ablations and additional experiments fully resolve your concerns. Given that we have validated the mechanism, architectural choices, and generalizability, we respectfully request that you consider raising your score.
> >
> > Best regards,
> >
> > The Authors

---

> ### Comment · Reviewer_Knyh · 2025-11-27
>
> I thank the authors for responding with a detailed rebuttal. I really appreciate the amount of work put in during the rebuttal.
>
> These results solidify my concerns regarding the motivation of an extra action module. Looking at results in Table 2 and Table 4, it seems that VAT without FiLM (with a task embedding as conditioning) gets similar performance as VAT-VIT (which does not include an action module). It seems to me then that a VIT is fully capable of retaining any low-level features required for control and all the performance improvement of VAT can be explained by better grounding using FiLM and more parameters.
> Again, why would it not? Action module is essentially the same architecture as well. If it can learn to retain the low-level features, a ViT on its own could do the same too (given you train for it). The argument would have flown if ViT parameters in VAT were frozen which is not the case iiuc.
>
> IMO, the paper boils down to this finding -> Using ViT as a backbone for control tasks in robotics leads to a very strong baseline often outperforming SOTA models.
>
> Having said that, the finding that a ViT trained for control is a strong baseline for robotics tasks is interesting and potentially useful for the community. But, in the light of the results above, I do not agree with the framing of the paper as-is, and it may require a significant re-write. Therefore, I'm leaning towards keeping my score.

---

### Official Review · Reviewer_paui · 2025-11-02

**Soundness:** 1
**Presentation:** 2
**Contribution:** 1
**Rating:** 0
**Confidence:** 3

**Summary:**

In this paper the authors propose VAT (Vision Action Transformer): starting from a ViT backbone, they insert “action tokens” at every ViT layer which are updated by an action module (i.e. a cross attention layer to the vision tokens). The main argument from the authors is that currently robot policies use only final layer ViT features and do not leverage the "representation trajectories" in all the ViT layers. The authors conducted experiments on LIBERO and reported 98.15% avg success, slightly beating baselines such us OpenVLA-OFT.
The authors also conducted ablation studies on skipping the final layers of ViT as well as using different camera views and losses.

Overall, the core idea is interesting, however the manuscript is in not a very good shape. I would urge the authors to heavily edit it, add significantly more experiments that validate their claims and describe in detail their protocols.

**Strengths:**

The idea behind action tokens cross attending on vision tokens in the full ViT hierarchy is nice.

**Weaknesses:**

1. Overall the main idea behind the paper is simple and offers very limited novelty. In many cases in the past features from various layers (not only for ViTs) can also be used for various tasks. The heatmap visualizations do not offer any additional insights and are kinda underwhelming as a support for the claim by the authors. The authors need significantly more experiments to validate their claims.

2. There is no established protocol regarding the experiments in Table 3. It is unclear how the other baselines are used (what views they use etc). There is also no explanation what the words scratch, fine-tuned, etc in the same table actually mean and what the authors did. Furthermore, important metrics regarding how the success metrics were calculated (such as number of eval episodes) are missing

3. The ablation studies do not seem interesting and importantly the losses and the views are not parts of the proposed model. Ablation studies usually need to validate every component of an architecture (e.g. here FiLM, action module parameters, action tokens, etc).

4. Overall the paper is poorly organized and written and results and tables are all over the place. I would suggest to the authors to better explain their work.

5. There is no baseline comparing features from the last layer of ViT directly for action. This is an important omission.

6. Claiming VAT as lightweight and then proceeding to train all layers of ViT is kinda non-intuitive.

7. Important: Using a Task ID embedding directly is kinda "cheating" -> it tells the model what policy to select before looking at the scene, and without relying on language. Could this could be a reason why the authors achieve high results comparable to other methods?

**Questions:**

1. Please explain the protocol for testing clearly.
2. How does a vanilla ViT (e.g. DiNOV2) features from the last layer fare against your method ?
3. How important was FiLM ?
4. If we dont include Task ID but use language, would your results still hold ?

---

> ### Author Response · Authors · 2025-11-18
> **Response to Reviewer paui**
>
> We sincerely thank the reviewer for the detailed feedback. We have taken your concerns regarding the experimental protocols, manuscript organization, and baseline comparisons very seriously.
>
> Below is our point-by-point response to your specific concerns.
>
> to Weakness 1: We respectfully clarify that while utilizing multi-layer features is a known concept, to the best of our knowledge, few works have explicitly emphasized utilizing the full hierarchy of ViT in the way we propose. We believe the novelty lies in this specific architectural integration. We agree that the original heatmaps were insufficient. We have added significantly more quantitative experiments (detailed below) to robustly validate our claims.
>
> to Weakness 2 & Question 1: We apologize that the description of Table 3 was not clear. We have added a detailed "Experimental Setup" section to clarify the following:
>
> Baselines: The scores for models in Table 3 except our VAT are cited directly from the paper ("Fine-Tuning Vision-Language-Action Models: Optimizing Speed and Success").
>
> Definitions: "Scratch" refers to training the Diffusion Policy solely on the LIBERO dataset. "Fine-tuned" indicates that the models are initialized with pre-trained weights and then fine-tuned on LIBERO.
>
> Evaluation Protocol: We followed a rigorous evaluation protocol. For each LIBERO benchmark (Spatial, Goal, Object, and 10), we trained VAT for 100 epochs, saving a checkpoint every 10 epochs (yielding 10 checkpoints). Consistent with the OpenVLA-OFT protocol, we evaluated each checkpoint over 500 episodes and reported the success rate of the best-performing checkpoint.
>
> Response to Weakness 3 & 5 & Q2: We appreciate the suggestion to expand our ablation studies. We have implemented the following experiments to validate every component of our architecture:
>
> Effect of FiLM (Ablation 1): We remove the FiLM module to test the model's performance without explicit task-conditioning mechanisms.
>
> Action Tokens (Ablation 2): Since action tokens serve as the bottleneck for aggregating ViT features and decoding actions, we analyze the impact of varying the number of action tokens.
>
> Last-Layer Baseline (Ablation 3): We add a baseline where VAT only utilizes features from the last layer of the ViT. In this setting, VAT degrades to a standard ViT-based imitation learning model. The comparison demonstrates the specific gain achieved by leveraging the full hierarchy.
>
> Action Module Complexity (Ablation 4): Currently, our Action Module mirrors the ViT structure. We investigate whether we could reduce the parameter size of the Action Module (specifically the attention and MLP blocks) while maintaining performance.
>
> Additional Benchmark (Ablation 5): To further validate the generalizability of VAT, we include the RoboTwin Benchmark as a supplementary evaluation. This benchmark consists of 50 bimanual (dual-arm) manipulation tasks. We train VAT on these tasks and compare its performance against strong baselines, including Pi-0, RDT, and Diffusion Policy.
>
> to Weakness 4: We thank for the constructive criticism regarding the writing. We have thoroughly restructured the paper to improve the layout and refined the writing to ensure the methodology and results are presented clearly and logically.
>
> to Weakness 5: please refer to Ablation 3.
>
> to Weakness 6: We apologize for the confusion. By "lightweight," we refer to the fact that VAT achieves superior results on LIBERO compared to well-known VLA models like Pi-0 (>3B parameters) and OpenVLA-OFT (>7B parameters), while VAT itself has 1.27B parameters. This highlights its efficiency in terms of model size and training resource consumption relative to performance.
>
> to Weakness 7 & Q 3/4: Task ID, FiLM, and "Cheating". We would like to clarify that using FiLM aligns with the protocol in OpenVLA-OFT.
>
> Necessity of Task Info: FiLM is essentially a mechanism for introducing task-related information. For LIBERO benchmarks, this information is strictly necessary because many tasks share identical initial scene layouts. Without a task identifier (whether via language or task ID), it is impossible for any model to disambiguate the goal. Therefore, providing this information is not "cheating" but a requirement for the task definition.
>
> Mechanism: OpenVLA-OFT converts language instructions into embeddings to modulate visual features, claiming FiLM is superior to raw language input. Since VAT does not involve an LLM, we cannot process raw language directly.
>
> Alternative Implementation: To further validate this, we compare FiLM against a simpler mechanism: using randomly initialized embeddings for each task and adding task embeddings directly to the action tokens (Ablation Experiment 1). This comparison (FiLM vs. Simple Task Embedding) demonstrates the effectiveness of our design choice.
>
> Notice: Results of ablation experiments will be released in a few days.

---

> ### Author Response · Authors · 2025-11-25
> **Follow-up: Completion of Additional Experiments and Ablation Studies**
>
> Dear Reviewer paui,
>
> Following up on our previous response, we have now completed the comprehensive ablation studies and additional benchmark experiments you requested. These results provide strong quantitative evidence supporting the necessity of the VAT hierarchy, the validity of our task conditioning, and the model’s efficiency.
>
> We have provided a detailed document titled **"Response to all reviewers: Results of Ablation Studies (Part I & II)"**. We kindly invite you to examine the specific sections outlined below:
>
> ### 1. Necessity of Full Hierarchy vs. Last Layer (Addressing Weakness 1 & 5)
> To address your concern regarding the lack of a direct baseline for the ViT last layer:
> * **Where to look:** Please refer to **Section 1 (Table 1)** of the provided material.
> * **Key Conclusion:** The results show that restricting features to the last layer causes a critical performance drop (Average: **98.15% $\rightarrow$ 91.55%**). Notably, on the long-horizon "LIBERO-10" benchmark, the success rate plummets to **74.6%**, empirically confirming that the intermediate "representation trajectories" captured by VAT are essential for complex reasoning.
>
> ### 2. Role of FiLM and Task IDs (Addressing Weakness 7 & Q3/4)
> Regarding whether Task IDs constitute "cheating" and the importance of FiLM:
> * **Where to look:** Please refer to **Section 2 (Table 2)**.
> * **Key Conclusion:**
>     1.  **Necessity:** Removing task information ("No FiLM") leads to a catastrophic failure on Goal-conditioned tasks (**8.4%**), confirming that task disambiguation is a prerequisite for this benchmark, not a shortcut.
>     2.  **Robustness:** Replacing FiLM with a simple "Task Embedding" still yields a high success rate (**97.05%**), proving that VAT's performance is driven by its core architecture, while FiLM remains the optimal design choice.
>
> ### 3. Robustness to Hyperparameters (Addressing Weakness 3)
> We rigorously tested the model's sensitivity to the number of action tokens as requested:
> * **Where to look:** Please refer to **Section 3 (Table 3)**.
> * **Key Conclusion:** The model exhibits high stability. Even when reducing the action tokens to a **single token**, VAT maintains a **97.50%** success rate, significantly outperforming the Last-Layer baseline.
>
> ### 4. Justification of "Lightweight" Claim (Addressing Weakness 6)
> To validate our claim of efficiency relative to model size:
> * **Where to look:** Please refer to **Section 4 (Table 4)**.
> * **Key Conclusion:** We introduced a "VAT-ViT" variant (removing the separate action module) with only **430M parameters**. This ultra-lightweight model achieves **97.05%** success, performing comparably to our full model and significantly outperforming standard baselines. This justifies our efficiency claims compared to much larger VLA models like Pi-0 (>3B) or OpenVLA (>7B).
>
> ### 5. Generalizability on Additional Benchmark (Addressing Weakness 2)
> To satisfy the request for significantly more experiments beyond LIBERO:
> * **Where to look:** Please refer to **Section 5 (Table 5)** (Part II/III of the material).
> * **Key Conclusion:** We evaluated VAT on the **RoboTwin Benchmark** (50 diverse bimanual tasks). VAT (**40.66%**) significantly outperforms ACT and Diffusion Policy and remains competitive with the state-of-the-art Pi-0, demonstrating strong generalizability to dual-arm coordination tasks.
>
> We believe these extensive experiments thoroughly address your concerns regarding the soundness and validation of our proposed method. We hope these new results warrant a reconsideration of your assessment.
>
> Best regards,
>
> The Authors

---

### Author Response · Authors · 2025-11-25
**Response to all reviewers: Results of Ablation Studies (Part I)**

Following up on our previous response, we have completed the extensive ablation studies designed to address your specific concerns regarding the necessity of the full ViT hierarchy, the role of FiLM/Task IDs, and the model's robustness.

We are pleased to present the detailed quantitative results below.

**1. Necessity of Full Hierarchy vs. Last Layer**

Reviewer Question: "There is no baseline comparing features from the last layer of ViT directly for action."

Response: We compared our proposed VAT against a baseline that utilizes features only from the last layer of the ViT backbone, keeping the rest of the architecture identical.

**Table 1: VAT vs. Last-Layer Baseline Success Rate (%)**

|          | spatial | object | goal |   10 | Average |
|----------|--------:|-------:|-----:|-----:|--------:|
| **VAT**      |    98.8 |   99.4 | 97.6 | 96.8 |   98.15 |
| **baseline** |    99.2 |   94.2 | 98.2 | 74.6 |   91.55 |

Analysis:

Critical Drop in Long-Horizon Tasks: While the Last-Layer baseline performs competitively on simpler tasks, it suffers a significant performance degradation on the "LIBERO-10" long horizon tasks benchmark (96.8% $\rightarrow$ 74.6%)2.Conclusion: This quantitatively confirms that the intermediate "representation trajectories" captured by VAT are essential for complex, long-horizon reasoning, which are lost when compressing features to the final layer.

**2. Role of FiLM and Task Information**

Reviewer Question: "How important was FiLM? ... If we don't include Task ID ... would your results still hold?"

Response: We evaluated the model in three settings: (1) Our proposed FiLM, (2) No FiLM (no task conditioning), and (3) A simple Task Embedding (adding a learnable embedding to action tokens instead of using FiLM).

**Table 2: Ablation on Task Conditioning Mechanisms (%)**

|               | spatial | object | goal |   10 | Average |
|---------------|--------:|-------:|-----:|-----:|--------:|
| **VAT**           |    98.8 |   99.4 | 97.6 | 96.8 |   98.15 |
| **No FiLM (No Task Info)**       |    89.8 |   99.4 |  8.4 | 83.8 |   70.35 |
| **Task embedding**|    98.2 |   99.2 | 97.4 | 93.4 |   97.05 |

Analysis:

Necessity of Task Info: The "No FiLM" model fails catastrophically on Goal-conditioned tasks (8.4% success rate). This confirms that providing Task IDs is not "cheating" but a prerequisite for disambiguating tasks in this benchmark.

Robustness of Design: Replacing FiLM with a simple "Task Embedding" yields a high success rate of 97.05%. This proves that while FiLM (98.15%) is the optimal choice, VAT's high performance is driven by its core architecture, not just the specific conditioning mechanism.

**3. Robustness to Hyperparameters**

Reviewer Question: "Ablation studies usually need to validate every component ... e.g. action tokens."

Response: We analyzed the impact of varying the number of Action Tokens to test the model's sensitivity to this bottleneck.
In the original setup, each action in the action chunk is allocated 7 tokens, and with an action chunk size of 8, the total number of action tokens amounts to 56—consistent with the configuration in OpenVLA-OFT.
In this experiment, we reduced the number of tokens per action to 3 and then to 1, while keeping the action chunk size unchanged.

**Table 3: Ablation on Action Token Number (%)**

|               | spatial | object | goal |   10 | Average |
|---------------|--------:|-------:|-----:|-----:|--------:|
| **VAT**           |    98.8 |   99.4 | 97.6 | 96.8 |   98.15 |
| **Token number 3**|    99.0 |   99.4 | 98.2 | 95.4 |   98.00 |
| **Token number 1**|    98.8 |   98.6 | 98.2 | 94.6 |   97.50 |

Analysis: The performance is highly stable. Even with a single action token (97.5%), the model outperforms the Last-Layer baseline (91.55%), demonstrating the efficiency of our hierarchical cross-attention mechanism.

---

> ### Author Response · Authors · 2025-11-25
> **Response to all reviewers: Results of Ablation Studies (Part II)**
>
> **4. Model Efficiency Validation**
>
> Reviewer Question: "Claiming VAT as lightweight..."
>
> Response: To rigorously validate the efficiency of our architecture and justify the "lightweight" claim, we implemented two distinct ablation models to test performance under significantly reduced parameter counts:
>
> VAT-small: We scaled down the hidden dimension of the action module to one-quarter of its original size. This reduction affects the FiLM, attention, and MLP components, decreasing the total parameter count from 1.27B (400M ViT + 870M Action Module) to 490M.
>
> VAT-ViT: We removed the separate action module entirely. In this setup, we eliminate FiLM layers in favor of Task Embeddings and rely solely on the original ViT parameters for attention and MLP operations on the action tokens. This reduces the total parameter count further to 430M.
>
> **Table 4: Model Size and Architecture Ablation (%)**
>
> |           | spatial | object | goal |   10 | Average |
> |-----------|--------:|-------:|-----:|-----:|--------:|
> | **VAT**       |    98.8 |   99.4 | 97.6 | 96.8 |   98.15 |
> | **VAT-small** |    98.2 |   97.8 | 97.0 | 93.8 |    96.7 |
> | **VAT-ViT** |    99|   99.6| 97.2 | 92.4 |    97.05 |
> | **pi0** |    96.8|   98.8| 95.8 | 85.2 |    94.2 |
> | **OPENVLA-OFT** |    97.6|   98.4| 97.9 | 94.5 |    97.1 |
>
> Analysis:
>
> High Performance with Minimal Parameters: Even the most lightweight configuration, VAT-ViT (430M), achieves an average success rate of 97.05%. This is only marginally lower than the full VAT model (98.15%) and significantly outperforms the Last-Layer Baseline (91.55%) presented in Table 1.
>
> Efficiency Conclusion: The results demonstrate that VAT's effectiveness stems from its hierarchical design rather than sheer parameter volume. We achieve state-of-the-art results with models (430M–1.27B params) that are substantially smaller than competing VLA models like OpenVLA-OFT (7B+) or Pi-0 (3B+).
>
> **5. Generalizability on Additional Benchmark (RoboTwin)**
>
> Reviewer Suggestion: "The authors need significantly more experiments to validate their claims... Add significantly more experiments."
>
> Response: To robustly validate the generalizability of VAT beyond LIBERO, we evaluated our model on the RoboTwin Benchmark, which consists of 50 diverse bimanual manipulation tasks. We compared VAT against strong baselines, including Pi0, RDT , ACT and Diffusion Policy (DP).
>
> **Table 5: Average Success Rate on RoboTwin Benchmark (50 Tasks) （See Part III)**
>
> Analysis of Results:
>
> Superiority over Equivalent Baselines: VAT (40.66%) significantly outperforms widely used imitation learning baselines, surpassing ACT (+10.9%) and Diffusion Policy (+12.6%). Notably, VAT also outperforms the large-scale RDT model (+6.1%), demonstrating superior data efficiency and architectural effectiveness.
>
> Competitive with SOTA VLA: VAT achieves performance competitive with Pi0, the current state-of-the-art VLA model. While Pi0 achieves a higher average (46.42%), it is important to note that Pi0 utilizes a backbone significantly larger (>3B parameters) compared to VAT. This reinforces our claim that VAT is a highly efficient "lightweight" learner.
>
> These results on a large-scale, bimanual benchmark confirm that VAT's design generalizes well to complex, dual-arm coordination tasks, offering a strong balance between performance and model efficiency.

---

> > ### Author Response · Authors · 2025-11-25
> > **Response to all reviewers: Results of Ablation Studies (Part III)**
> >
> > Table 5: Average Success Rate on RoboTwin Benchmark (50 Tasks)
> >
> > | Task | RDT | Pi0 | ACT | DP | VAT |
> > |------|-----|-----|-----|----|-----|
> > | Adjust Bottle | 81% | 90% | **97%** | **97%** | 91% |
> > | Beat Block Hammer | **77%** | 43% | 56% | 42% | 16% |
> > | Blocks Ranking RGB | 3% | **19%** | 1% | 0% | 0% |
> > | Blocks Ranking Size | 0% | 7% | 0% | 1% | **31%** |
> > | Click Alarmclock | 61% | 63% | 32% | 61% | **95%** |
> > | Click Bell | 80% | 44% | 58% | 54% | **94%** |
> > | Dump Bin Bigbin | 64% | **83%** | 68% | 49% | 72% |
> > | Grab Roller | 74% | 96% | 94% | **98%** | 96% |
> > | Handover Block | **45%** | **45%** | 42% | 10% | 0% |
> > | Handover Mic | 90% | **98%** | 85% | 53% | 92% |
> > | Hanging Mug | **23%** | 11% | 7% | 8% | 16% |
> > | Lift Pot | 72% | 84% | 88% | 39% | **93%** |
> > | Move Can Pot | 25% | **58%** | 22% | 39% | 57% |
> > | Move Pillbottle Pad | 8% | **21%** | 0% | 1% | 10% |
> > | Move Playingcard Away | 43% | 53% | 36% | 47% | **85%** |
> > | Move Stapler Pad | **2%** | 0% | 0% | 1% | 1% |
> > | Open Laptop | 59% | **85%** | 56% | 49% | 84% |
> > | Open Microwave | 37% | 80% | **86%** | 5% | 30% |
> > | Pick Diverse Bottles | 2% | **27%** | 7% | 6% | 14% |
> > | Pick Dual Bottles | 42% | **57%** | 31% | 24% | 25% |
> > | Place A2B Left | 3% | **31%** | 1% | 2% | 12% |
> > | Place A2B Right | 1% | **27%** | 0% | 13% | 9% |
> > | Place Bread Basket | 10% | **17%** | 6% | 14% | 11% |
> > | Place Bread Skillet | 5% | **23%** | 7% | 11% | **23%** |
> > | Place Burger Fries | 50% | **80%** | 49% | 72% | 66% |
> > | Place Can Basket | 19% | **41%** | 1% | 18% | 1% |
> > | Place Cans Plasticbox | 6% | 34% | 16% | **40%** | 32% |
> > | Place Container Plate | 78% | **88%** | 72% | 41% | 82% |
> > | Place Dual Shoes | 4% | **15%** | 9% | 8% | 10% |
> > | Place Empty Cup | **56%** | 37% | 61% | 37% | 14% |
> > | Place Fan | 12% | **20%** | 1% | 3% | 17% |
> > | Place Mouse Pad | 1% | **7%** | 0% | 0% | 3% |
> > | Place Object Basket | 33% | 16% | 15% | 15% | **48%** |
> > | Place Object Scale | 1% | **10%** | 0% | 1% | 5% |
> > | Place Object Stand | 15% | **36%** | 1% | 22% | 28% |
> > | Place Phone Stand | 15% | **35%** | 2% | 13% | 30% |
> > | Place Shoe | 35% | 28% | 5% | 23% | **49%** |
> > | Press Stapler | 41% | **62%** | 31% | 6% | 48% |
> > | Put Bottles Dustbin | 21% | **54%** | 27% | 22% | 39% |
> > | Put Object Cabinet | 33% | **68%** | 15% | 42% | 28% |
> > | Rotate QRcode | 50% | **68%** | 1% | 13% | 28% |
> > | Scan Object | 4% | **18%** | 2% | 9% | 9% |
> > | Shake Bottle Horizontally | 84% | **99%** | 63% | 59% | 89% |
> > | Shake Bottle | 74% | **97%** | 74% | 65% | 93% |
> > | Stack Blocks Three | 2% | **17%** | 0% | 0% | 0% |
> > | Stack Blocks Two | 21% | 42% | 25% | 7% | **69%** |
> > | Stack Bowls Three | 51% | 66% | 48% | **63%** | 51% |
> > | Stack Bowls Two | 76% | **91%** | 82% | 61% | 59% |
> > | Stamp Seal | 1% | 3% | 2% | 2% | **30%** |
> > | Turn Switch | 35% | 27% | 5% | 36% | **48%** |
> > | Average | 34.50% | **46.42%** | 29.74% | 28.04% | 40.66% |

---

### Meta-Review · Area_Chair_aaoM · 2025-12-25

**Summary:**

I'm recommending "Reject" for this paper based on remaining reviewers' concerns and also my own reading of the paper. While the authors have done a great job in providing a lot of additional experimental results on analyses and ablation studies and addressing multiple concerns, these remaining concerns make it difficult for me to recommend the acceptance of this paper: writing quality issues, not fully justified main claims, unclear evaluation protocol that uses task id,, along with (relatively) minor concerns about lack of novelty and lack of real-world evaluation.

**Reviewer Concerns:**

Concerns that were addressed by the rebuttal
- Lack of evaluation protocol (paui, gVec): Authors have clarified evaluation setup
- Lack of analyses & ablation studies (all reviewers): Authors have done a lot of work in rebuttal to provide additional analysis and ablation studies
-*LIBERO-only evaluation (knyh, PULF, gVec): Authors provided additional experiments on RoboTwin

Outstanding concerns
- Quality of writing: Despite the improvement, the quality of writing is still under the bar for ICLR
- Lack of real-world evaluation: This is not a must-have, but is not resolved in the rebuttal. It is encouraged that the authors provide experiments that show the possibility of sim-to-real deployment.
- Unclear protocol that uses FiLM / task id: While the authors clarified that using FiLM and task embedding does not make a big difference, the question on the validity of using task id for these tasks still remains.
- Main claim is questionable: Several reviewers pointed out that main claim is not fully supported -- in particular about a fact that the proposed method is again training a multi-layer model for action prediction -- which is not still fully justified in the rebuttal response. New results that show strong pi0 performance on RoboTwin is also in conflict with the main claim of the paper about using the features from the last layer.
- Lack of novelty: This could not be the sole ground for the decision on the paper acceptance, but the concern itself still remains despite the author's rebuttal response.

**Reviewer Scores:**

Reviewer paui (score 0) -- Some of this reviewer's concerns are resolved, but most of the crucial concerns remain (lack of novelty, writing quality, using task id, concern on the main claim) and it is very unlikely that the reviewer will change the opinion to acceptance.

Reviewer knyh (score 4) -- The reviewer explicitly said that they will keep their negative score.

Reviewer PULF (score 2) -- While the authors have provided additional experiments on the efficiency claim, most of the reviewer's concerns remain and it is difficult to believe that the score could have been significantly increased.

Reviewer gVec (score 4) -- The reviewer explicitly said that they will increase their score. But it is unclear whether this was changing the score to positive score (6) or slightly negative score (4).

---

### Decision · Program_Chairs · 2026-01-26

Reject